# VISUAL PLANNING: LET'S THINK ONLY WITH IMAGES

**Yi Xu**[1*]  **Chengzu Li**[1*]  **Han Zhou**[1*]  **Xingchen Wan**[2]  **Caiqi Zhang**[1]
**Anna Korhonen**[1]  **Ivan Vulić**[1]

[1]Language Technology Lab, University of Cambridge    [2]Google

`{yx465, cl917, hz416, cz391, alk23, iv250}@cam.ac.uk`
`xingchenw@google.com`

## ABSTRACT

Recent advancements in Large Language Models (LLMs) and their multimodal extensions (MLLMs) have substantially enhanced machine reasoning across diverse tasks. However, these models predominantly rely on pure text as the medium for both expressing and structuring reasoning, even when visual information is present. In this work, we argue that language may not always be the most natural or effective modality for reasoning, particularly in tasks involving spatial and geometrical information. Motivated by this, we propose a new paradigm, Visual Planning, which enables planning through purely visual representations for these "vision-first" tasks, as a supplementary channel to language-based reasoning. In this paradigm, planning is executed via sequences of images that encode step-by-step inference in the visual domain, akin to how humans sketch or visualize future actions. We introduce a novel reinforcement learning framework, Visual Planning via Reinforcement Learning (VPRL), empowered by GRPO for post-training large vision models, leading to substantial improvements in planning in a selection of representative visual navigation tasks, FROZENLAKE, MAZE, and MINIBEHAVIOR. Our visual planning paradigm outperforms all other planning variants that conduct reasoning in the text-only space. Our results establish Visual Planning as a viable and promising supplement to language-based reasoning, opening new avenues for tasks that benefit from intuitive, image-based inference. Code is available at: `https://github.com/yix8/VisualPlanning`.

## 1 INTRODUCTION

Large Language Models (LLMs) (Brown et al., 2020; Ouyang et al., 2022; Anil et al., 2023) have demonstrated strong capabilities in language understanding and generation, as well as growing competence in complex reasoning, enabled by their chain-of-thought reasoning abilities (Wei et al., 2022b). Building on these advances, recent work extends LLMs to support multiple modalities, yielding so-called Multimodal Large Language Models (MLLMs) (Reid et al., 2024; Hurst et al., 2024): they incorporate visual embedded information at the input to tackle a broader spectrum of tasks, such as visual spatial reasoning (Liu et al., 2023; Li et al., 2024a) and navigation (Gu et al., 2022; Li et al., 2024b). However, despite their multimodal inputs, these methods perform reasoning purely in the text format during inference, from captioning visual content (Hao et al., 2025) to generating verbal rationales (Zhang et al., 2024b).

Building on this observation, we argue that performing multimodal reasoning only in the text pathway may not always offer the most intuitive or effective strategy, particularly for tasks that depend heavily on visual information and/or are 'vision-first' by design. Indeed, recent results from multimodal benchmarks (Roberts et al., 2025; Li et al., 2024a; Chen et al., 2024; Cheng et al., 2025) offer growing evidence that purely language-based reasoning falls short in certain domains, particularly those involving spatial, geometric, or physical dynamics (Zhang et al., 2025a; Li et al., 2025a). Such reliance on grounding visual information into text before reasoning introduces a modality gap that hinders the model's ability to capture visual features and state transitions. This highlights a potential

---

*Equal contribution.

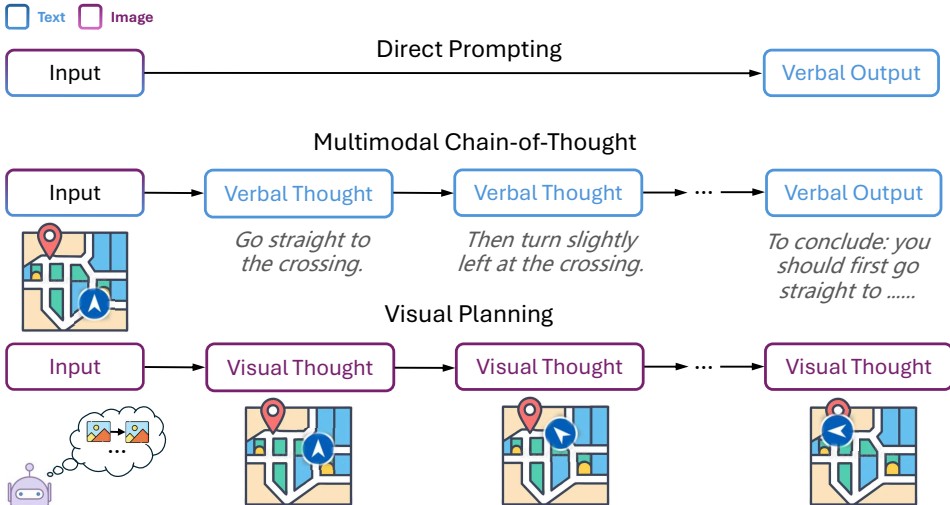

Figure 1: Comparison of reasoning paradigms. The traditional approaches (*top* and *middle* rows) generate verbose and inaccurate textual plan, while the Visual Planning paradigm (*bottom* row) predicts the next visual state directly, forming a pure image trajectory without language mediation.

shortcoming of current MLLMs: while they process image inputs, they do not naturally "*think*" in images. For instance, tasks such as planning a route through a maze, designing the layout of a room, or predicting the next state of a mechanical system are often better served by visual representations, as verbal descriptions may be less effective and struggle to accurately capture complex spatial reasoning relationships. These examples suggest a broader question, which we aim to tackle in this work: *can models directly plan in non-verbal modalities, such as images, without being mediated by text?*

Cognitive science also offers compelling motivation for this question (Moulton & Kosslyn, 2009). Dual Coding Theory (Paivio, 1991) proposes that human cognition operates through both verbal and nonverbal channels, each capable of independent representational and inferential processes. Recent work on MLLMs incorporates interleaved text and images as reasoning steps (Hu et al., 2024; Li et al., 2025b; Zhang et al., 2025b). However, they still remain fundamentally text-driven and rely on tool-based visualizations as auxiliary information for reasoning traces, with reasoning still mainly embedded in verbal traces. For instance, Visual Sketchpad (Hu et al., 2024) employs external tools to generate sketches as visual aids, and MVoT (Li et al., 2025b) generates per-step visualizations from language-based actions but still reasons in text for decision-making. As such, a truly visual-only reasoning paradigm that avoids any text-based reasoning proxies remains underexplored.

In this work, we propose a new paradigm, *Visual Planning*, where reasoning is structured as a sequence of images, but without the mediation of language. To the best of our knowledge, this is the first attempt to investigate whether models can achieve planning purely through visual representations. Rather than generating textual rationales and answers, our approach produces step-by-step visualizations that encode planning or inference steps directly in images. As a pioneering exploration, it circumvents the modality mismatch that occurs when visual problems must be forced into explanations in verbal form, reinforces state transitions, and provides a new trackable interface for tasks like navigation (Li et al., 2024a), and visual problem-solving (Hao et al., 2025; Zhang et al., 2026).

Specifically, we explore this paradigm using the Large Vision Model (LVM) (Bai et al., 2024) trained exclusively on images and video frames with **no** textual data. This design choice removes potential confounders introduced by language-based supervision and enables a clean investigation of whether models can reason purely within the visual modality. Motivated by the success of reinforcement learning in acquiring reasoning capabilities within the language modality (Guo et al., 2025a) and its strong generalization performance (Chu et al., 2025), we propose Visual Planning via Reinforcement Learning (VPRL), a novel two-stage reinforcement learning framework empowered by GRPO (Shao et al., 2024) for visual planning. It involves a distinct initializing stage for encouraging the exploration of the policy model in the given environment, which is then followed by reinforcement learning with a progress reward function.

We validate the feasibility of our paradigms on grid-based navigation as a representative of spatial planning tasks, including MAZE (Ivanitskiy et al., 2023), FROZENLAKE (Wu et al., 2024b), and MINIBEHAVIOR (Jin et al., 2023), where one agent is requested to navigate to a target location successfully without violating environment constraints. Our experiments reveal that the visual planning paradigm substantially surpasses the traditional textual reasoning method by supervised fine-tuning (SFT), achieving 27% higher average exact-match rate. In addition to better performance, our novel method VPRL exhibits stronger generalization to out-of-distribution scenarios than the SFT method in the visual planning paradigm (VPFT). To the best of our knowledge, we are the first to apply RL to image generation in the context of planning, with contributions as follows:

- We propose a new reasoning paradigm, *Visual Planning*, and validate the feasibility of visual reasoning without any use of text and language for reasoning.

- We introduce VPRL, a novel two-stage training framework that applies RL to achieve visual planning via sequential image generation.

- We demonstrate empirically that VPRL significantly outperforms the traditional textual reasoning paradigm and supervised baselines in visual spatial planning settings, achieving substantial gains in task performance and exhibiting improved generalization.

## 2 VISUAL PLANNING VIA REINFORCEMENT LEARNING

### 2.1 THE VISUAL PLANNING PARADIGM

The majority of prior visual reasoning benchmarks (Goyal et al., 2017; Akula et al., 2021; Yue et al., 2024) can be and is typically tackled by grounding the visual information in the textual domain (Gurari et al., 2018; Peng et al., 2024; Zhang et al., 2024a), followed by a few steps of textual reasoning. However, once the visual content is mapped to text (e.g., object names, attributes, or relations), the problem gets reduced to a language reasoning task, where the reasoning is carried out by the language model, even without reflecting any information from the visual modality.

Our visual planning paradigm is fundamentally different. It performs planning purely within the visual modality as a holistic process, where the actions are not explicitly predicted but instead implicitly represented by transitions between visual states. We formally define visual planning as a process of generating a sequence of intermediate images $\hat{\mathcal{T}} = (\hat{v}_1, \ldots, \hat{v}_n)$, where each $\hat{v}_i$ represents a visual state that together constitute a visual planing trajectory, given the input image $v_0$. Specifically, let $\pi_\theta$ denote a generative vision model parameterized by $\theta$. The visual planning trajectory $\hat{\mathcal{T}}$ is generated autoregressively, where each intermediate visual state $\hat{v}_i$ is sampled conditioned on the initial state and previously generated states:

$$\hat{v}_i \sim \pi_\theta(v_i | v_0, \hat{v}_1, ..., \hat{v}_{i-1}). \tag{1}$$

### 2.2 REINFORCEMENT LEARNING FOR LARGE VISION MODELS

Reinforcement learning (RL) has shown notable advantages in improving the generalization of autoregressive models by optimizing with *sequence-level* rewards beyond token-level supervision signals (Chu et al., 2025). In autoregressive image generation, an image is represented as a *sequence of visual tokens*. Inspired by the success of RL in language reasoning (Guo et al., 2025a), we introduce an RL-based training framework for visual planning empowered by Group Relative Policy Optimization (GRPO) (Shao et al., 2024). It leverages the transitions between visual states to compute the reward signals while verifying the constraints from the environments. To enforce the policy model that generates valid actions with diverse exploration during the RL process, we then propose a novel two-stage reinforcement learning framework for visual planning. In Stage 1, we first apply supervised learning to initialize the policy model with random trajectories. Model's visual planning is then optimized by the RL training in Stage 2.

**Stage 1: Policy Initialization.** In this stage, we initialize the model $\pi_\theta$ by training it on random trajectories obtained by random walks in the environment. The goal here is to generate valid sequences of visual states and retain exploration capability in a 'simulated' environment. For training, each

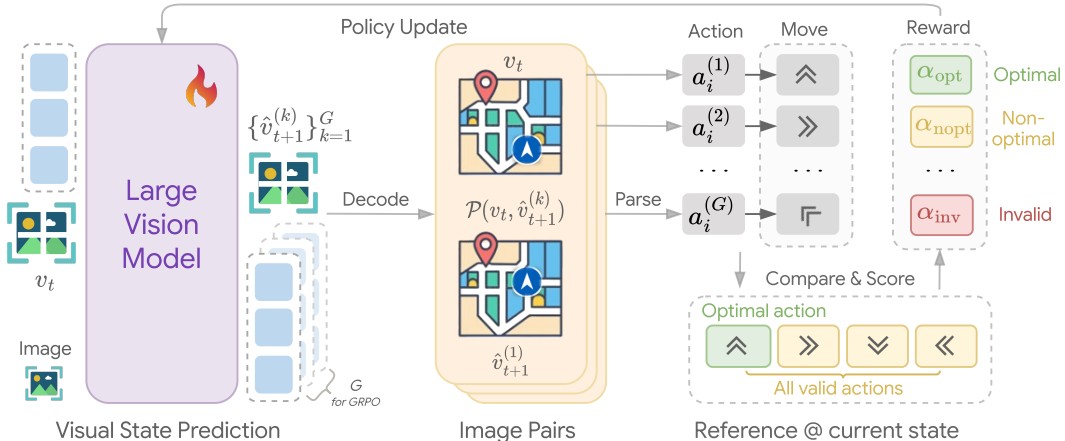

Figure 2: An overview of the proposed VPRL framework, illustrated with autoregressive large vision models for image generation in the context of a visual navigation task. We train the visual policy model with GRPO, using the *progress* reward that encourages progressing actions and penalizes invalid actions, yielding goal-aligned visual planning.

trajectory $\mathcal{T}$ consists of a sequence of visual states $(v_0, \ldots, v_n)$. From each trajectory, we extract $n-1$ image pairs of the form $(v_{\leq i}, v_{i+1})$, where $v_{\leq i}$ represents the prefix sequence $(v_0, \ldots, v_i)$. Given an input prefix $v_{\leq i}$, to prevent overfitting to the specific transition and encourage stochasticity, we randomly sample one candidate state $\tilde{v}_{i+1}$ from all possible valid next states as the supervision target, and minimize the following loss function of visual planning via fine-tuning (**VPFT**):

$$\mathcal{L}_{\text{VPFT}}(\theta) = -\mathbb{E}_{(v_{\leq i}, \tilde{v}_{i+1})}\Big[\log \pi_\theta\big(\tilde{v}_{i+1} \,\big|\, v_{\leq i}\big)\Big]. \tag{2}$$

Overall, the first stage serves as a warm-up for subsequent optimization, focusing on producing visually coherent outputs and enhancing the generation quality.

**Stage 2: Reinforcement Learning for Visual Planning.** Building on Stage 1, where the model is initialized with random trajectories, it acquires the effective exploration capability. This property is essential for RL, as it ensures coverage over all possible transitions and prevents collapse to suboptimal behaviors. Stage 2 then leverages this ability to simulate the outcomes of potential actions by generating the next visual state and guiding the model to effectively do the planning. During this stage, the RL algorithm provides feedback and rewards based on the correctness of the simulated actions, gradually enabling the model to learn effective visual planning.

Specifically, given an input prefix $v_{\leq i}$, the behavior model $\pi_{\theta_{\text{old}}}$ samples a group of $G$ candidate responses $\{\hat{v}_{i+1}^{(1)}, \ldots, \hat{v}_{i+1}^{(G)}\}$. The candidate response is then scored using a composite reward function $r(v_i, \hat{v}_{i+1}^{(k)})$, which quantifies whether the generated visual state represents meaningful progress toward the goal state. The reward design and implementations are described in detail in the next paragraph.

Instead of relying on a learned critic to estimate value functions which may introduce additional sources of uncertainty and complexity, GRPO provides more computationally efficient and interpretable training signals by computing relative advantages through comparisons within the group. In this case, the relative advantage of each candidate is $A^{(k)} = \frac{r^{(k)} - \text{mean}\left\{r^{(1)}, r^{(2)}, \ldots, r^{(G)}\right\}}{\text{std}\left\{r^{(1)}, r^{(2)}, \ldots, r^{(G)}\right\}}$. To guide the model toward producing responses with higher advantages, we update the policy $\pi_\theta$ by maximizing the following objective:

$$\mathcal{J}_{\text{VPRL}}(\theta) = \mathbb{E}_{v_{\leq i} \sim \mathcal{D}, \{\hat{v}_{i+1}^{(k)}\}_{k=1}^{G} \sim \pi_{\theta_{\text{old}}}(\cdot|v_{\leq i})}$$
$$\left[\frac{1}{G}\sum_{i=1}^{G}\min\left(\rho^{(k)}A^{(k)}, \ \text{clip}\left(\rho^{(k)}, 1-\epsilon, 1+\epsilon\right)A^{(k)}\right) - \beta\, D_{\text{KL}}\left(\pi_\theta \,\|\, \pi_{\text{ref}}\right)\right], \tag{3}$$

where $\mathcal{D}$ is the prefix distribution and $\rho^{(k)} = \frac{\pi_\theta(\hat{v}_{i+1}^{(k)}|v_{\leq i})}{\pi_{\theta_{\text{old}}}(\hat{v}_{i+1}^{(k)}|v_{\leq i})}$ is the importance sampling ratio.

**Reward Design.** Unlike discrete actions or text tokens, visual outputs are sparse, high-dimensional, and not easily decomposable into interpretable units. In our visual planning framework, the challenge is even more specific: whether the generated visual state can correctly reflect the intended planning action. Consequently, our reward design emphasizes both adherence to environment constraints (validity of state transitions) and progress toward the goal.

Formally, let $\mathcal{A}$ denote the set of *valid* actions and $\mathcal{E}$ the set of *invalid* ones (e.g., violations of physical constraints or hallucinated new entities in the environment). To interpret and evaluate the intended action that connects the current state $v_i$ to a generated candidate state $\hat{v}_{i+t}^{(k)}$, we introduce *1)* the *dynamics interpreter* $\mathcal{D} : \mathcal{V} \times \mathcal{V} \to \mathcal{A} \cup \mathcal{E}$ to parse the transition and *2)* the *progress estimator* $P : \mathcal{V} \to \mathbb{N}$ to quantify the progress.

The dynamics interpreter $\mathcal{D}$ evaluates the transitions $a \in \mathcal{A} \cup \mathcal{E}$ for validity, which, by implementation, can be a dynamics model (Qiu et al., 2025) or a rule-based system to elicit actions from state pairs, or a neural model as holistic validator that judges transitions without explicitly inferring actions. The progress estimator $P(v)$ quantifies progress by estimating the remaining steps or effort required to reach the goal from each visual state. By comparing the agent's current and predicted state, we partition the generated candidate states $\mathcal{A} \cup \mathcal{E}$ into three disjoint subsets:

$$\mathcal{A}_{\mathrm{opt}} = \big\{ a \in \mathcal{A} : P(\hat{v}_{i+1}^{(k)}) < P(v_i) \big\}, \quad \mathcal{A}_{\mathrm{nopt}} = \big\{ a \in \mathcal{A} : P(\hat{v}_{i+1}^{(k)}) \geq P(v_i) \big\}, \quad \mathcal{E}_{\mathrm{inv}} = \mathcal{E}.$$

Here, $\mathcal{A}_{\mathrm{opt}}$ corresponds to optimal actions that reduce the distance to the goal, $\mathcal{A}_{\mathrm{nopt}}$ captures non-optimal but still valid actions, and $\mathcal{E}_{\mathrm{inv}}$ denotes invalid ones determined by the dynamics interpreter.

Based on this partition, we define the *progress reward* function $r(v_i, \hat{v}_{i+1}^{(k)})$:

$$\underbrace{\alpha_{\mathrm{opt}} \cdot \mathbb{I}\big[\mathcal{D}(v_i, \hat{v}_{i+1}^{(k)}) \in \mathcal{A}_{\mathrm{opt}}\big]}_{\text{optimal}} + \underbrace{\alpha_{\mathrm{nopt}} \cdot \mathbb{I}\big[\mathcal{D}(v_i, \hat{v}_{i+1}^{(k)}) \in \mathcal{A}_{\mathrm{nopt}}\big]}_{\text{non-optimal}} + \underbrace{\alpha_{\mathrm{inv}} \cdot \mathbb{I}\big[\mathcal{D}(v_i, \hat{v}_{i+1}^{(k)}) \in \mathcal{E}_{\mathrm{inv}}\big]}_{\text{invalid}}, \quad (4)$$

where $\alpha_{\mathrm{opt}}, \alpha_{\mathrm{nopt}}, \alpha_{\mathrm{inv}}$ are reward coefficients. In our experiments, we set $\alpha_{\mathrm{opt}} = 1$, $\alpha_{\mathrm{nopt}} = 0$, and $\alpha_{\mathrm{inv}} = -5$, thereby rewarding progressing actions, assigning zero to non-progressing actions, and heavily penalizing invalid transitions.

## 3 EXPERIMENTS AND RESULTS

**Tasks.** To evaluate our proposed visual planning paradigm, we select representative tasks where planning can be expressed and executed entirely in the visual modality. We focus on tasks where state transitions are visually observable, distinguishing them from language-centric tasks like code generation (Lai et al., 2023) or traditional visual question answering. This design allows us to analyze planning behavior without relying on textual rationales or symbolic outputs. To compare visual planning with language-based reasoning, we experiment with 3 visual navigation environments: FROZENLAKE (Wu et al., 2024b), MAZE (Ivanitskiy et al., 2023), and MINIBEHAVIOR (Jin et al., 2023). All of them can be solved in both modalities, which enables a direct parallel comparison of pros and cons between visual planning and language reasoning strategies.

- FROZENLAKE: It is initially proposed by Wu et al. (2024b) and implemented with Gym (Brockman, 2016). It simulates a grid-based frozen lake, where the agent is supposed to start from the designated position and find its way to the destination safely without falling into the 'holes'.
- MAZE: Given an initial image describing the maze layout, the model is supposed to go through the maze from the starting point (green point) to the destination (red flag).
- MINIBEHAVIOR: The agent is first required to reach the printer from the starting point and pick it up. After that, the agent should go to the table and drop the printer. This task consists of 2 additional actions, including 'pick' and 'drop'.

We construct synthetic datasets for the tasks with varying levels of complexity in patterns and environments. Details on data collection and implementation are provided in Appendix E.1.

**Models.** To explore visual planning without any language influence as confounders and enables a clean investigation, we select models trained exclusively on visual data without any exposure to

Table 1: Performance of the closed- and open-source models on FROZENLAKE, MAZE, and MINIBEHAVIOR. VPRL performs consistently the best (**bold**) across all tasks. [†] denotes the post-trained model. $\mathbb{A}$ represents texts and 🖼 represents images. The last column AVG. reports the average performance across three tasks.

| Model | Input | Output | FROZENLAKE | | MAZE | | MINIBEHAVIOR | | AVG. | |
|---|---|---|---|---|---|---|---|---|---|---|
| | | | EM (%) | PR (%) | EM (%) | PR (%) | EM (%) | PR (%) | EM (%) | PR (%) |
| Closed-Source Model | | | | | | | | | | |
| Gemini 2.0 Flash | | | | | | | | | | |
| - Direct | $\mathbb{A}$+🖼 | $\mathbb{A}$ | 21.2 | 47.6 | 8.3 | 31.4 | 0.7 | 29.8 | 10.1 | 36.3 |
| - CoT | $\mathbb{A}$+🖼 | $\mathbb{A}$ | 27.6 | 52.5 | 6.9 | 29.8 | 4.0 | 31.2 | 12.8 | 37.8 |
| Gemini 2.5 Pro (*think*) | $\mathbb{A}$+🖼 | $\mathbb{A}$ | 72.0 | 85.0 | 21.5 | 35.5 | 37.6 | 59.9 | 43.7 | 60.1 |
| Open-Source Model | | | | | | | | | | |
| Qwen 2.5-VL-Instruct-7B | | | | | | | | | | |
| - Direct | $\mathbb{A}$+🖼 | $\mathbb{A}$ | 1.2 | 15.0 | 0.6 | 14.5 | 0.3 | 9.8 | 0.7 | 13.1 |
| - CoT | $\mathbb{A}$+🖼 | $\mathbb{A}$ | 8.2 | 29.1 | 2.3 | 15.2 | 0.5 | 14.7 | 3.7 | 19.7 |
| - SFT[†] | $\mathbb{A}$+🖼 | $\mathbb{A}$ | 68.6 | 84.4 | 60.9 | 70.3 | 31.3 | 56.1 | 53.6 | 69.9 |
| LVM-7B | | | | | | | | | | |
| - VPFT[†] (ours) | 🖼 | 🖼 | 75.4 | 79.5 | 59.0 | 64.0 | 33.8 | 52.2 | 56.1 | 65.2 |
| - VPRL[†] (ours) | 🖼 | 🖼 | **91.6** | **93.2** | **74.5** | **77.6** | **75.8** | **83.8** | **80.6** | **84.9** |

textual data during pretraining. For visual planning, we use the Large Vision Model (LVM-7B) (Bai et al., 2024) as the backbone, which is only trained on image sequences and videos. We train the model with 1) supervised fine-tuning over golden planning trajectory (**VPFT**) and 2) two-stage reinforcement learning (**VPRL**), resulting in two system variants with visual planning. For RL training, we start with a rule-based parsing function as the dynamics interpreter to parse the image pairs to actions, and a heuristic progress estimator, with details enclosed in Appendix E.3.

For baselines, to facilitate parallel comparison for language-based planning, we adopt Qwen 2.5-VL-Instruct (Bai et al., 2025), on both inference-only (Direct[1] and CoT) and post-training settings (SFT and RL), trained on the same data as the visual planner. We further evaluate multimodal reasoning performance of proprietary models with Gemini 2.0 Flash (Kampf & Brichtova, 2025) and advanced thinking model Gemini 2.5 Pro (Gemini, 2025). Full training details, model versions, and hyperparameters are provided in Appendix E.4.

**Evaluation Metrics.** We adopt two complementary evaluation metrics for the selected tasks. Let $\mathcal{O} = \{\mathcal{T}^{(1)}, \mathcal{T}^{(2)}, \ldots, \mathcal{T}^{(M)}\}$ denote the set of all shortest optimal trajectories of length $n$, where each trajectory is $\mathcal{T}^{(m)} = (v_1^{(m)}, \ldots, v_n^{(m)})$, and let $\hat{\mathcal{T}} = (\hat{v}_1, \ldots, \hat{v}_n)$ denote the predicted trajectory.

- **Exact Match (EM)** is defined as $\text{EM} = \max_{m \in \{1, \ldots, M\}} \prod_{j=1}^{n} \mathbb{I}(\hat{v}_j = v_j^{(m)})$, evaluating whether $\hat{\mathcal{T}}$ coincides with any $\mathcal{T}^{(m)} \in \mathcal{O}$. EM requires the entire trajectory to be valid and of minimal length, and accepts all optimal solutions rather than a single reference. Here, the equality $\hat{v}_j = v_j^{(m)}$ refers to whether the two states can be reached from their respective previous states by applying the same action. This means that the comparison is made at the level of environment transitions rather than a pixel-wise match between images. In other words, two states are treated as the same if they represent the same underlying configuration, even when their pixel values are not identical.

- **Progress Rate (PR)** is defined as $\text{PR} = \max_{m \in \{1, \ldots, M\}} \frac{1}{n} \sum_{j=1}^{n} \left[ \prod_{k=1}^{j} \mathbb{I}(\hat{v}_k = v_k^{(m)}) \right]$, measuring the ratio of consecutive correct steps (valid forward moves) from the start that align with at least one optimal trajectory. PR thus provides a softer signal than EM, capturing the model's ability to make meaningful progress towards a full solution. The same state equality is applied as in EM.

**Textual planning falls short in both proprietary models and open-sourced tuning baselines.** Table 1 shows that proprietary models yield average EM below 50% and PR only marginally above 50% at best, underscoring the challenges these tasks pose for current models despite being intuitive for humans. On the other hand, while task-specific training provides partial improvement, the overall

---

[1]Direct denotes answer prediction without being instructed to conduct intermediate reasoning.

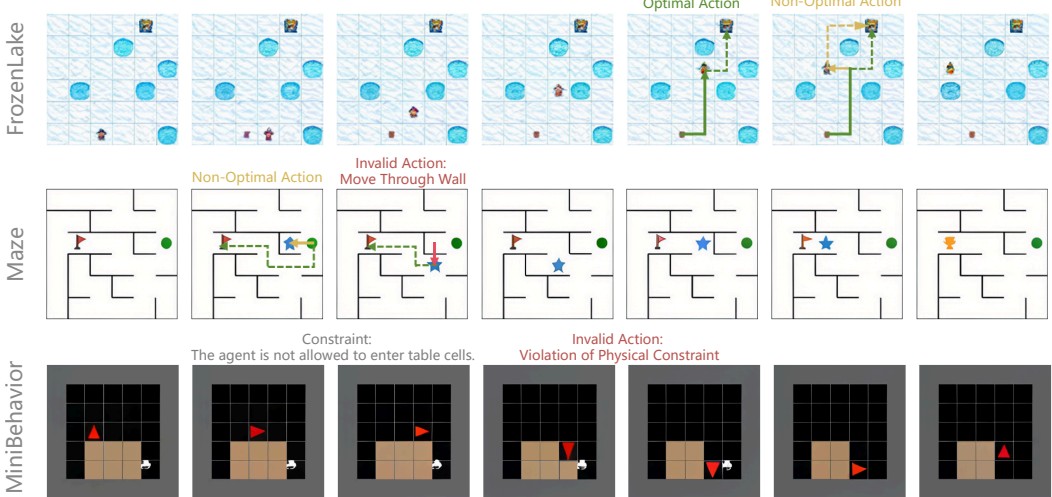

Figure 3: Illustration of each task with generated visual planning traces from LVM, covering different types of actions (optimal, non-optimal and invalid). More cases can be found in Appendix F.6.

performance of fine-tuned textual planners remains unsatisfactory, through either directly generating planned actions (SFT in Table 1) or first captioning the image with different textual representations and then generating answers (Table 2). We also observe that, unlike the notable gains of RL in the pure language domain (Guo et al., 2025a), RL yields limited performance gains when applied to text-based planning with multimodal inputs. Table 2 shows that when using progress reward as in VPRL or directly using the Progress Rate metric as the outcome reward, none of the variants surpasses the SFT baseline. We attribute the bottleneck of language-based planning with SFT and RL to the modality gap, which leads to inaccuracies in grounding visual information into text and thereby constrains performance. Further discussion is provided in Section 4.

**Visual planning achieves better performance than textual baselines via RL.** While supervised fine-tuning (VPFT) achieves performance comparable to text-based SFT, it remains constrained by imitation and limited exposure to diverse trajectories. By contrast, our two-stage reinforcement learning framework (VPRL) substantially improves the planning capability, achieving the strongest overall results. After Stage 2 optimization, the model attains near-perfect accuracy on FROZENLAKE (91.6% EM, 93.2% PR) and maintains strong performance on more complex MAZE and MINIBEHAVIOR tasks, outperforming VPFT by over 20% on average. As expected, the

Table 2: Performance of text-based planning variants on FROZENLAKE. See Table 7 in Appendix F.2 for the full results.

| Model | EM (%) | PR (%) |
|---|---|---|
| Qwen 2.5-VL-Instruct-7B | | |
| - SFT | | |
|     - *Direct* | 68.6 | **84.4** |
|     - *w/ Coordinates* | **74.4** | 82.7 |
|     - *w/ ASCII* | 73.1 | 83.4 |
| - GRPO | | |
|     - *w/ VPRL progress reward* | 54.4 | 69.9 |
|     - *w/ PR metric reward* | 60.1 | 74.3 |

improvement is fully driven by outcome-based optimization in Stage 2, as Stage 1 alone yields near-random behavior (Table 10 in Appendix F.6). Unlike VPFT, which mainly fits the training distribution, VPRL enables exploration of diverse actions and learning from their consequences through reward-driven updates, thereby capturing underlying planning rules and achieving stronger performance.

**VPRL shows robustness with scaling complexity.** The advantage of RL also holds when we study the performance of different methods with respect to task difficulties, where a larger grid usually relates to higher difficulties. In Figure 5, as the grid size increases from $3 \times 3$ to $6 \times 6$ in the FROZENLAKE environment, Gemini 2.5 Pro's EM score drops sharply from 98.0% to 38.8%. In comparison, our visual planners not only maintain higher accuracy at all grid sizes but also exhibit a much flatter performance curve. Similarly, VPRL demonstrates even greater stability than VPFT, with EM remaining at 97.6% on $3 \times 3$ grids and still achieving 82.4% on $6 \times 6$, indicating strong robustness. We observe similar trends in other tasks; see Appendix F.3 for other tasks.

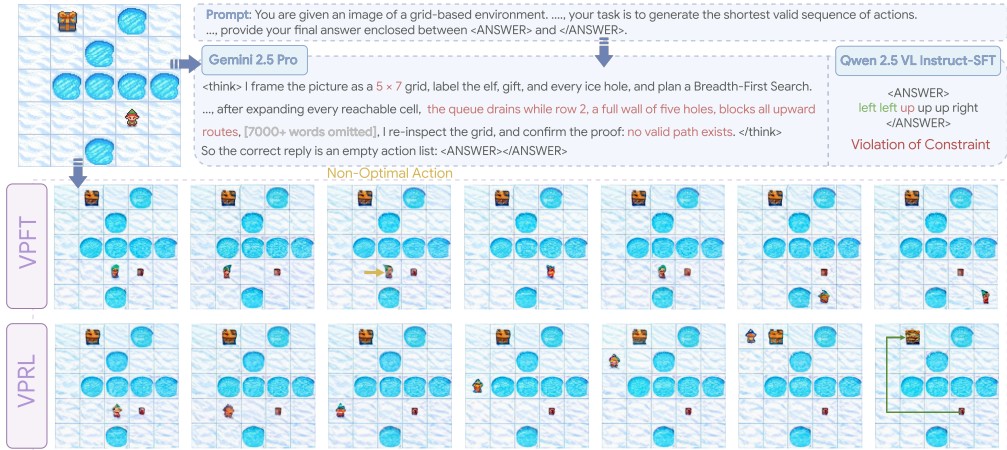

Figure 4: Visualization of a test example from FROZENLAKE comparing visual planning variants (VPFT and VPRL) with language-based reasoning variants.

## 4 DISCUSSIONS AND ANALYSIS

**Error Analysis and Case Study.** We conduct error analysis for language-based planning and visual planning. We observe that textual planning systems with both SFT and RL are prone to errors when grounding visual inputs to verbalized descriptions during the inference process, with 25.7% of generated coordinate-based layout descriptions and 22.3% of generated ASCII-based representations being mismatched with ground-truth layouts. Qualitative analysis of response from textual RL baselines (Figure 9 in Appendix F.2.1) and proprietary models (Figure 4) also reveal similar observations. Taken together, these results demonstrate an inherent modality gap where language may not be the most accurate and effective representation for vision-first problem. For visual planning, Figure 3 presents visual planning traces generated by LVM across different tasks. We observe that the model occasionally takes non-optimal actions that deviate from the shortest path (FROZENLAKE example). Surprisingly, VPRL demonstrates the ability to take detours to bypass the obstacles while still progressing towards the goal, whereas VPFT lacks this flexibility and gets stuck, as shown in Figure 4. Additional traces covering optimal, non-optimal, and invalid cases can be found in Appendix F.6. Beyond these in-domain analyses, we further evaluate generalization on larger unseen grids and perturbed image inputs, with results reported in Appendix F.4.

**Random policy initialization enables exploration.** We ablate whether we could directly use VPFT as the policy model for GRPO training rather than intentionally initialize a model with random trajectories. We hypothesize that VPFT, trained via teacher-forcing, inherently limits exploration by repeatedly generating similar actions, resulting in identical rewards. In this case, it yields zero advantage, preventing policy updates and hindering effective learning. We empirically validate this hypothesis by comparing the exploration capabilities of VPFT with VPRL Stage 1 (Figure 6). We observe that VPFT's entropy rapidly declines throughout training, eventually approaching zero, indicating severe exploration limitations. Although earlier VPFT checkpoints exhibit higher entropy, they produce significantly more invalid actions. In contrast, VPRL Stage 1 demonstrates significantly higher entropy, closely approaching the entropy of the uniform random planner, while maintaining a lower invalid action ratio, justifying the necessity of Stage 1 random initialization for RL framework.

**VPRL reduces invalid action failure.** Another important benefit of VPRL lies in its effectiveness in reducing invalid actions. To quantify this, we analyze all failed trajectories and compute the proportion that contains at least one invalid action, as opposed to failures caused by non-optimal but valid plans. We refer to this as the *invalid-failure* ratio. As shown in Table 6, VPFT exhibits a high ratio ranging from 61% to 78% over three tasks, while VPRL reduces this ratio by at least 24% in all cases, demonstrating that VPRL not only improves success rates, but also encourage the model to stay within valid action spaces during planning.

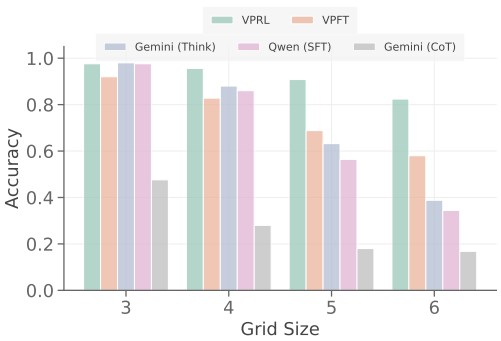 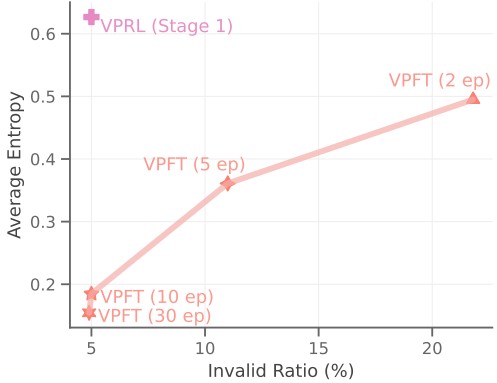

Figure 5: Evaluation of model performance on FROZENLAKE under varying levels of difficulty. As the environment complexity increases with larger grid sizes, language-based reasoning methods experience a sharp decline in performance, whereas visual planning methods exhibit a more gradual drop, demonstrating greater robustness.

Figure 6: Comparison of exploration capabilities between VPFT and VPRL Stage 1 on FROZEN-LAKE. VPRL Stage 1 achieves significantly better exploration efficiency, balancing high entropy with a low invalid action ratio, whereas VPFT struggles with diminishing entropy and increased invalid actions over training.

## 5 RELATED WORK

**MLLM Reasoning.** Recent work has extended CoT prompting (Wei et al., 2022c) to MLLMs through approaches such as grounding visual inputs into symbolic representations, such as graphs or bounding boxes (Zhang et al., 2024b; Lei et al., 2024). Other approaches integrate tools to generate visualizations during reasoning (Hu et al., 2024; Zhou et al., 2024). For example, o3 model (OpenAI, 2025) incorporates visual rationales using tools such as cropping and zooming. MVoT (Li et al., 2025b) is also essentially a form of tool use: instead of relying on external modules, it invokes itself to generate visualizations of textual reasoning. These methods primarily conduct reasoning in language, with visual components merely illustrating the textual rationale rather than serving as the medium of reasoning. In this work, we take a step further to explore whether multi-step planning can emerge purely within visual representations, enabling reasoning without relying on language at all.

**Reinforcement Learning for Visual Reasoning.** Reinforcement learning has been applied across a wide range of vision-related tasks, especially given the rise of GRPO as in DeepSeek-R1 (Guo et al., 2025a). Concurrently, in object detection, visual perception (Yu et al., 2025) is optimized though rewarding high Intersection-over-Union (IoU) scores between predicted and ground-truth bounding boxes (Shen et al., 2025). For visual reasoning tasks such as Visual Question Answering (VQA), GRPO has been utilized to optimize the models for longer, more coherent, and logically grounded reasoning traces in textual responses (Liu et al., 2025; Zhou et al., 2025; Zhang et al., 2025c; Team et al., 2025). More recently, similar methods have also been applied to image generation tasks for recursive refinement with textual instructions (Guo et al., 2025b; Wang et al., 2025; Jiang et al., 2025). These approaches focus on pixel-level fidelity and semantic alignment with text, whereas our work leverages RL for goal-oriented visual planning, optimizing multi-step decision-making through visual state transitions without any reliance on language.

**Action-conditional Generative Models.** Action-conditional generative models has focused on constructing latent representations of the world and predicting future observations conditioned on given actions (Ha & Schmidhuber, 2018; Ball et al., 2025). These models learn transition dynamics and are central to model-based reinforcement learning, where they allow agents to simulate potential outcomes without interacting directly with the environment (Hafner et al., 2019). While effective for representation learning and short-horizon prediction, action-conditional generative models do not perform planning and must therefore be coupled with an external planner. In contrast, our approach constitutes a holistic planner that internalizes planning within the visual generative flow, which is more effective for visual tasks than traditional text-based planners that suffer from a modality gap. It can also benefit from action-conditional generative models by using predicted observations as inputs.

## 6 CONCLUSION

In this work, we present Visual Planning as a new paradigm for reasoning in visually oriented tasks, challenging the prevailing reliance on language as the primary medium for structured inference. By enabling models to operate entirely through visual state transitions without textual mediation, we show that purely visual representations provide performance comparable to text-based planning in spatially grounded and dynamic tasks, establishing visual planning as a viable alternative. More importantly, our proposed two-stage reinforcement learning framework, VPRL, empowered by GRPO, further enhances the planning capabilities of large vision models. It obtains significant gains across three visual navigation tasks, achieving 27% EM improvements in task performance than language-based planning and showing stronger generalization on out-of-distribution scenarios. These findings underscore the promise of visual planning as a powerful alternative to text-based approaches. We believe our work opens up a rich new direction for multimodal research, offering a foundation for building more intuitive, flexible, and powerful reasoning systems across a wide range of domains.

## ACKNOWLEDGEMENTS

The work has been supported by the UK Research and Innovation (UKRI) Frontier Research Grant EP/Y031350/1 (the UK government's funding guarantee for ERC Advanced Grants) awarded to Anna Korhonen at the University of Cambridge. The work has also been supported in part by a Royal Society University Research Fellowship (no 221137; 2022-) awarded to Ivan Vulić, and by the UK EPSRC grant EP/T02450X/1.

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

## A   THE USE OF LARGE LANGUAGE MODELS

Large language models (LLMs) were used as general-purpose tools in this work. Specifically, LLMs assisted in polishing the writing to improve clarity and readability.

## B   ETHICS STATEMENT

Our research adheres to rigorous ethical guidelines. We verified the licenses of all softwares and datasets we used in this study and ensured full compliance with their terms. Furthermore, we have thoroughly assessed the project and do not anticipate any additional potential risks.

## C   REPRODUCIBILITY STATEMENT

Appendix E.1 introduces the datasets in details with statistics and processing procedure. Appendix E.2 introduces models we used in our paper, and Appendix E.3 provides detailed information regarding reward implementation for VPRL method. All hyper-parameters and training details are listed in Appendix E.4 for reproducibility. Appendix E.5 introduces the licences for the data and models we used. Prompting templates are shown in Appendix G. All data and scripts will be released publicly upon acceptance to facilitate reproducibility.

## D   LIMITATIONS AND FUTURE WORK

In this work, we focus exclusively on Large Vision Model (LVM) to investigate visual planning capabilities by eliminating language as a confounding factor *for research purposes*. As such, this choice constraints the model size to 7B as the only available size of LVM, and excludes recently released native multimodal models capable of generating multimodal outputs (Chern et al., 2024; Wu et al., 2024a). However, we argue that the visual planning paradigm can be extended to broader multimodal generation models for use in more diverse tasks, combined with more modalities, as long as they support image (visual) generation (Li et al., 2026).

Additionally, explicitly generating images introduces computational overhead during inference compared to a textual response. However, we argue that language-based reasoning performs worse than visual planning and can be equally or even more time-consuming, especially for thinking models (Gemini, 2025). In our demonstration, Gemini generated over 7,000 thinking tokens yet failed to provide the correct answer in the end. The computation overhead introduced by image generation can be alleviated through more compact image representations using fewer tokens (Choudhury et al., 2024), which we advocate for future research.

Another limitation in this work lies in the implementation of dynamics interpreter. For simplicity, we adopt the rule-based approach that compares pixel-wise features between the current state and the previous state (details in Appendix E.3). While effective in our controlled setup, broader task settings involving more complex visual structures are yet to be explored. Nevertheless, we argue that the underlying reward formulation remains extensible, but the primary challenge lies in defining reliable progress signals as visual transitions become more complex. Such signals could be supported by either dynamic models that elicit actions from pairs of images (e.g. segmentation (Ravi et al., 2024) or contour detection (Linsley* et al., 2020)) or a holistic neural model (e.g. Gemini (Gemini, 2025) or a learned reward model) that directly judges whether the transitions are valid without explicitly inferring actions. Alternatively, trajectory-level rollouts with final success feedback could be leveraged to identify actions that contribute to progress toward successful outcomes, eliminating the requirement for an explicit dynamics interpreter. We encourage future research to explore more robust and scalable designs for interpreting visual transitions to advance visual planning systems.

**Broader Impact.** This work introduces a novel paradigm of visual planning, where agents reason and act entirely within the visual modality without reliance on textual intermediaries. By demonstrating that models can plan through sequences of images, this research opens new possibilities for the way human and AI system interacts, particularly in domains like robotics, navigation, and assistive technologies, where perception and decision-making are tightly coupled. As the first step toward planning grounded purely in visual representations, our work lays the foundation for AI systems that

integrate both verbal and non-verbal reasoning. We advocate for future research into more holistic multimodal thinking systems where interleaved text and image traces enable richer, more human-like reasoning, and emphasize the importance of strengthening the visual component in such traces for improved planning and cognition.

# E    IMPLEMENTATION DETAILS

## E.1    DATASET

**Task Action Space.** FROZENLAKE and MAZE both involve four primitive navigation actions: `up`, `down`, `left`, and `right`. MINIBEHAVIOR includes a more complex action space with two additional operations: `pick`, `drop`.

**Dataset preparation.** For both FROZENLAKE and MAZE, we construct environments of grid sizes ranging from $3 \times 3$ to $6 \times 6$. For each size, we sample 1250 environments, with 1000 used for training and 250 held out for testing (Table 3). Each environment here is guaranteed to have a unique layout, and the agent is randomly initialized at a grid from which the goal is reachable, forming the initial state $v_0$. Due to the relatively limited diversity of environments layout in MINIBEHAVIOR, where the complexity arises primarily from the action space, sampling unique environments in a small grid size becomes challenging. Therefore, we focus only on grid sizes $7 \times 7$ and $8 \times 8$, allowing duplicates in layout but varying agent spawn positions to ensure sufficient data volume. To prevent data leakage, we split the dataset based on layout identity, ensuring no layout overlap between the training and test sets.

We next describe the dataset construction procedures corresponding to the training setups outlined in Section 3, with the number of samples per task summarized in Table 4.

- **SFT in Text** (Baseline): For each environment, we sample an optimal trajectory consisting of a sequence of visual states $(v_0, \ldots, v_n)$ as the ground truth. Each transition between states is determined by an action, enabling us to derive a corresponding verbalized action sequence $(a_0, \ldots, a_{n-1})$. The input to the model is formulated by concatenating a textual prompt with an image representation of the initial state $v_0$, while the target output is the verbalized action sequence representing the optimal trajectory. We further ablate different variants of the baseline with various representations and tuning methods (SFT and RL) in Appendix F.2. The detailed prompts for all variants are provided in Appendix G.

- **VPFT**: We utilize the same set of optimal trajectories as the language-based reasoning baseline described above. In the visual scenario, each trajectory generates multiple input-target pairs by pairing the state at timestep $t$ as the input with the subsequent state at timestep $t + 1$ as the target.

- **VPRL**:
    - Stage 1: This dataset serves solely for format control training of the visual backbone. For each environment, we enumerate all possible trajectories from the initial state as $v_0$ and generate corresponding input-target pairs. Duplicate pairs are filtered to maintain a balanced distribution.
    - Stage 2: To ensure fairness and comparability, this dataset uses the same input states as VPFT.

- **VPFT***: We conduct an ablation study (indicated with *) where VPFT is also trained in two stages, mirroring the structure of VPRL. Stage 1 follows the same procedure as VPRL Stage 1, focusing on format supervision using enumerated visual inputs. Stage 2 reuses the original VPFT training pipeline, learning from optimal trajectories. Experimental results and analysis see Appendix F.5.

*Note:* For both textual and visual planning setups, evaluation is performed using only the initial state $v_0$ of each test environment as input.

**Dataset Statistics.** We evaluate the performance of different system variants in in-distribution and out-of-distribution (OOD) settings. Table 3 shows the training data distribution over different grid sizes across three tasks. The numbers of training and testing samples for different system variants are shown in Table 4. For OOD evaluation, the enlarged grid sizes are shown in Table 9. OOD evaluation data includes 250 samples for each task.

Table 3: Distribution of training dataset by grid sizes for each task. Value indicates the number of environments.

| FROZENLAKE | | | | |
|---|---|---|---|---|
| Grid Size | 3 | 4 | 5 | 6 |
| Train | 1000 | 1000 | 1000 | 1000 |
| Test | 250 | 250 | 250 | 250 |
| MAZE | | | | |
| Grid Size | 3 | 4 | 5 | 6 |
| Train | 1000 | 1000 | 1000 | 1000 |
| Test | 250 | 250 | 250 | 250 |
| MINIBEHAVIOR | | | | |
| Grid Size | 7 | | 8 | |
| Train | 796 | | 801 | |
| Test | 204 | | 199 | |

Table 4: Number of training and test samples for each task and method. For visual planning, the numbers here are represented in image pairs, which correspond to the same number of trajectories for SFT in Text.

| Task | Split | SFT in Text | VPFT | VPRL | | VPFT* | |
|---|---|---|---|---|---|---|---|
| | | | | Stage 1 | Stage 2 | Stage 1 | SFT |
| FROZENLAKE | Train | 4000 | 12806 | 170621 | 12806 | 170621 | 12806 |
| | Test | 1000 | 1000 | N/A | 1000 | N/A | 1000 |
| MAZE | Train | 4000 | 14459 | 156682 | 14459 | 156682 | 14459 |
| | Test | 1000 | 1000 | N/A | 1000 | N/A | 1000 |
| MINIBEHAVIOR | Train | 1597 | 9174 | 90808 | 9174 | 90808 | 9174 |
| | Test | 403 | 403 | N/A | 403 | N/A | 403 |

### E.2 MODELS

Large Vision Model (LVM) (Bai et al., 2024) is an autoregressive models for image generation, which is only pretrained with image sequences with no exposure to language data. The model uses a tokenizer based on the VQGAN architecture (Esser et al., 2021), which extracts visual information from raw images and encodes it into 256 tokens from a fixed codebook. The image is generated in an auto-regressive manner with discrete tokens, which are then fed into the image detokenizer. Although LVM supports multiple model sizes, only the 7B-parameter version is publicly available; thus, we use this variant in our experiments. For a fair comparison, we use Qwen 2.5-VL-Instruct (Bai et al., 2025) with a matching parameter size as our language-based baseline.

### E.3 REWARD IMPLEMENTATION

We adopt a rule-based state-action parsing function as the dynamics interpreter $\mathcal{D}$ and heuristic progress estimator $P$ in VPRL. For the progress estimator, we apply the Breadth First Search (BFS) to search for the optimal trajectories and calculate the progress at each position in the grid for each task, in order to obtain a progress map covering all positions. The progress map are then used as a reward signal to guide VPRL training.

Specifically, for state-action parsing function, we parse the state and identify the difference between the current state and the previous state through a pixel-wise feature extractor. We first convert both input and predicted states into a coordinate-based representation by dividing the image into a grid based on its size. Each region corresponds to a discrete coordinate in the environment. To reduce sensitivity to color and focus on structural differences, we convert all images to grayscale. We subsequently compute the Intersection-over-Union (IoU) between each coordinate in the predicted state and the coordinate in the input state that contains the player (input coordinate). The coordinate in the predicted state with the highest IoU is selected as the predicted agent position. The action is

Table 5: Hyper-parameters of training both textual and visual planners.

| Hyper-Parameters | SFT in Text | RL in Text | VPFT | VPRL | | VPFT* | |
|---|---|---|---|---|---|---|---|
| | | | | Stage 1 | Stage 2 | Stage 1 | SFT |
| Epochs | 30 | 10 | 30 | 10 | 10 | 10 | 30 |
| Learning Rate | 1e-5 | 1e-5 | 1.5e-4 | 1.5e-4 | 5e-5 | 1.5e-4 | 1.5e-4 |
| Train Batch Size | 4 | 1 | 8 | 8 | 1 | 8 | 8 |
| Group Size | N/A | 8 | N/A | N/A | 10 | N/A | N/A |
| Grad Accumulation | 1 | 1 | 1 | 1 | 1 | 1 | 1 |
| GPUs | 8 | 8 | 8 | 8 | 8 | 8 | 8 |

then inferred by comparing the start and predicted positions according to task-specific movement rules. For example, in the MAZE environment, movement across walls is not allowed and would be considered invalid.

Notably, to detect the invalid transitions, such as the disappearance of agents, we also calculate the pixel-wise mean squared error (MSE) between corresponding coordinates to measure local visual differences. If two coordinates exhibit significant MSE differences exceeding a predefined threshold, we treat them as the potential source and destination of a movement (agent disappears from one and appears in another). If only one such coordinate is found, we treat it as a disappearance event, indicating an invalid transition.

In MINIBEHAVIOR, we extend this logic to identify pick and drop actions. A pick is detected when the IoU between the printer's location in the input and predicted states falls below a threshold, indicating that the printer has been removed. A drop is inferred when a coordinate corresponding to the table region shows a large MSE increase, suggesting the printer has been placed there. Additional edge cases in these tasks are omitted for brevity.

For reward computation, if the predicted action is valid, we compare the progress values from the heuristic progress estimator $P$ between the input and predicted states. A reward of 1 is given if the predicted state shows greater progress toward the goal than the input state; otherwise, the reward is 0. Invalid actions are penalized with a reward of -5.

Our method and reward modeling approach are readily generalizable to other visual tasks. With reference to computer vision techniques such as segmentation (Ravi et al., 2024) and contour detection (Linsley* et al., 2020), the pixel-level analysis used in our framework can be easily extended to a wide range of structured visual environments. Furthermore, our reward design is broadly applicable to planning tasks in general. Since actions in most planning settings can naturally be categorized into one of three types (valid and helpful, valid but non-progressing, or invalid), our simple reward structure remains intuitive and effective across tasks.

### E.4 TRAINING DETAILS

In addition to VPRL, we include several training system variants as baselines that differ in supervision modalities (language vs. image) and optimization methods (SFT vs. RL), allowing us to compare language-based and vision-based planning while assessing the role of reinforcement learning.

**Visual Planning via Fine-Tuning (VPFT).** We propose Visual Planning via Fine-Tuning (VPFT) as a simplified variant of our framework, which shares the same training architecture as Stage 1 in Section 2.2, but replaces random trajectories with optimal planning trajectories. For each environment, we sample a distinct trajectory $(v_0^{opt}, v_1^{opt}, \ldots, v_n^{opt})$ representing the minimal-step path from the initial state $v_0^{opt} = v_0$ to the goal. At each step, the model is trained to predict the next state $v_{i+1}^{opt}$ given the prefix $v_{\leq i}^{opt}$. The objective is identical to Equation 2, with supervision from the optimal trajectory.

**Supervised Fine-Tuning (SFT) in Text.** In this baseline, planning is formulated in the language modality. Instead of generating an intermediate visual consequence of an action, the model produces a textual description of the intended action sequence. Formally, given an visual input state $v$ and a textual prompt $p$, which represents the task description, the model is trained to generate a verbalized action sequence $t = (t_1, \ldots, t_L)$, where each token $t_i \in \mathcal{V}_{text}$ represents an action. The input to

Table 6: We compute the percentage of failed trajectories that are caused by at least one invalid action, rather than a suboptimal but valid action. Lower values indicate better action validity control.

| Task | Invalid-Failure Ratio (%) | |
|------|------|------|
| | **VPRL** | VPFT |
| FROZENLAKE | **36.9** | 60.6 |
| MAZE | **25.1** | 73.7 |
| MINIBEHAVIOR | **29.6** | 78.3 |

the model is the concatenation of the prompt tokens and the visual tokens, and the target is the corresponding action sequence. Following prior work on supervised fine-tuning (SFT) (Wei et al., 2022a) in autoregressive models, we minimize the cross-entropy loss for action prediction:

$$\mathcal{L}_{\text{SFT}}(\theta) = -\mathbb{E}_{(v,t)} \left[ \sum_{i=1}^{L} \log \pi_\theta(t_i \mid t_{<i}, v, p) \right]. \tag{5}$$

Beyond directly training on action labels, we further conduct an ablation on FROZENLAKE with textual variants that verbalize the input state before predicting the action sequence. In particular, we explore two structured representations: **Coordinate** descriptions and **ASCII** grids. During training, the target sequence consists of the description tokens (encoding the environment layout in either coordinate or ASCII form) concatenated with the action labels that lead to the goal.

**Reinforcement Learning (RL) in Text.** We also extend RL to textual planning in the FROZENLAKE environment as an ablation. We optimize the textual planner with Group Relative Policy Optimization (GRPO). The reward design combines a fixed *format reward*, which enforces the correct output structure, with an outcome reward defined in two variants: (1) a progress-based reward identical to that used in VPRL, or (2) the Progress Rate (PR) metric directly used as the reward.

For all post-training experiments, we apply Low-Rank Adaptation (LoRA) (Hu et al., 2022) on both attention layers and feed-forward layers. The detailed hyper-parameters are shown in Table 5. Only the loss of the targets is calculated in an instruction-tuning manner (Wei et al., 2022a) for SFT. The image tokenizer and detokenizer are frozen during training. We use the AdamW optimizer Loshchilov & Hutter (2019) for all training procedures.

When SFT for textual planning and visual planning, we train the model for a maximum of 30 epochs. For VPRL, we first do stage 1 on random trajectories for 10 epochs for the purpose of exploration. We then use GRPO to optimize the model for planning for another 10 epochs for stage 2. We sample a group of 10 candidate responses per prompt to compute the advantages accordingly. To encourage a balance between exploration and exploitation, we apply a KL divergence penalty with a coefficient $\beta = 0.001$. For RL in the textual modality, we adopt the same 10 training epochs for fairness, with a group size of 8. We use the TRL library for training (von Werra et al., 2020). We've conducted our experiments on the machine with $8 \times$ A100 GPUs.

### E.5 LICENSES

Model-wise, Large Vision Model and Qwen 2.5 VL are under the Apache-2.0 license. TRL is under the Apache-2.0 license. We collect the MAZE dataset with our own Python scripts. FROZENLAKE is collected from OpenAI Gym under the MIT License.

## F RESULTS

### F.1 VPRL TRAINING

The reward curves with standard deviation for all tasks are shown in Figure 7. The shaded regions indicate the standard deviation across groups. For better visualization, we apply Gaussian smoothing to both the reward values and their corresponding standard deviations.

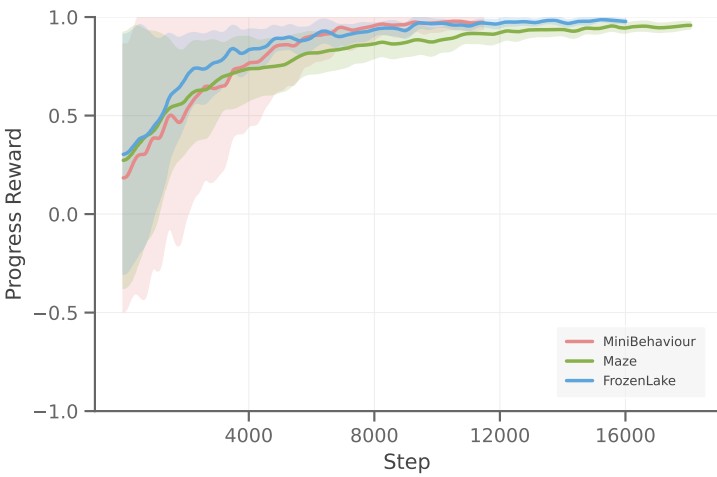

Figure 7: Reward curves with standard deviation for VPRL on FROZENLAKE, MAZE and MINIBEHAVIOR.

Table 7: Performance of text-based variants of Qwen-2.5-VL-Instruct-3B and 7B on FROZENLAKE. We report Exact Match (EM) and Progress Rate (PR) across all difficulty levels (L3–L6) and their average.

| Model | EM (%) | | | | | PR (%) | | | | |
|---|---|---|---|---|---|---|---|---|---|---|
| | L3 | L4 | L5 | L6 | Avg. | L3 | L4 | L5 | L6 | Avg. |
| Qwen 2.5-VL-Instruct-3B | | | | | | | | | | |
|   - SFT | | | | | | | | | | |
|     - *Direct* | 87.2 | 72.0 | 48.8 | 28.0 | 59.0 | 89.4 | 84.3 | 71.3 | 60.1 | 76.3 |
|     - *w/ Coordinates* | 87.2 | 78.0 | 64.8 | 30.8 | 65.2 | 89.6 | 82.5 | 74.0 | 57.2 | 75.8 |
|     - *w/ ASCII* | 79.6 | 75.6 | 58.8 | 34.0 | 62.0 | 83.3 | 82.5 | 74.8 | 59.1 | 74.9 |
|   - GRPO | | | | | | | | | | |
|     - *w/ VPRL progress reward* | 69.2 | 52.8 | 41.2 | 26.0 | 47.3 | 73.6 | 72.2 | 66.2 | 55.0 | 66.8 |
|     - *w/ PR metric reward* | 70.8 | 60.0 | 41.6 | 23.6 | 49.0 | 75.5 | 76.1 | 65.7 | 56.0 | 68.4 |
| Qwen 2.5-VL-Instruct-7B | | | | | | | | | | |
|   - SFT | | | | | | | | | | |
|     - *Direct* | 97.6 | 86.0 | 56.4 | 34.4 | 68.6 | 98.1 | 92.1 | 78.9 | 68.4 | 84.4 |
|     - *w/ Coordinates* | 93.2 | 88.0 | 74.8 | 41.6 | 74.4 | 94.1 | 89.7 | 81.5 | 65.5 | 82.7 |
|     - *w/ ASCII* | 93.2 | 86.0 | 68.0 | 45.2 | 73.1 | 94.1 | 88.6 | 81.3 | 69.6 | 83.4 |
|   - GRPO | | | | | | | | | | |
|     - *w/ VPRL progress reward* | 72.4 | 64.0 | 50.4 | 30.8 | 54.4 | 76.2 | 76.3 | 69.2 | 57.8 | 69.9 |
|     - *w/ PR metric reward* | 82.8 | 68.8 | 51.6 | 37.2 | 60.1 | 84.9 | 79.6 | 71.5 | 61.0 | 74.3 |
| LVM-7B | | | | | | | | | | |
|   - VPFT (ours) | 92.0 | 82.8 | 68.8 | 58.0 | 75.4 | 93.1 | 84.7 | 73.4 | 66.9 | 79.5 |
|   - **VPRL** (ours) | **97.6** | **95.6** | **90.8** | **82.4** | **91.6** | **98.4** | **96.0** | **93.0** | **85.6** | **93.2** |

## F.2 TRAINED TEXTUAL BASELINES AND REWARD DESIGN

To strengthen the comparison with our visual planners, we train different text-based baselines beyond the direct action-sequence SFT model reported in the main paper. We are interested in: 1) whether different textual representation influences the performance of language-based reasoning, and 2) whether reinforcement learning can help to improve the language-based planning performance with multimodal input.

**Trained SFT variants.** Specifically, we experiment with two alternative SFT variants that first describe the environment layout in different formats (coordinates and ASCII) before predicting the action sequence.

Table 8: Exact Match performance of VPFT and VPFT$^*$ across different grid sizes in FROZENLAKE.

| Model | Exact Match (%) | | | |
|---|---|---|---|---|
| | 3×3 | 4×4 | 5×5 | 6×6 |
| VPFT$^*$ | 86.4 | 73.6 | 50.0 | 33.2 |
| **VPFT** | **92.0** | **82.8** | **68.8** | **58.0** |

Table 9: Out-of-distribution (OOD) performance on enlarged grids. Models are trained on smaller grids and evaluated on the sizes indicated in parentheses.

| Model | FROZENLAKE (7×7) | | MAZE (7×7) | | MINIBEHAVIOR (9×9) | |
|---|---|---|---|---|---|---|
| | EM (%) | PR (%) | EM (%) | PR (%) | EM (%) | PR (%) |
| VPFT | 9.6 | 15.3 | 9.2 | 17.8 | 0.0 | 5.8 |
| **VPRL** | **20.4** | **31.2** | **10.0** | **21.6** | **0.4** | **14.7** |

- **SFT with Coordinates:** The model is trained to first output a coordinate-based description of the grid environment (e.g., positions of the agent, goal, and obstacles), followed by the full action sequence.
- **SFT with ASCII:** The model is trained to output an ASCII-based description of the environment layout before producing the action sequence. Specifically, S denotes the starting position, G the goal, H an ice hole, and F a passable cell.

The example input-output formats for different text-based reasoning variants are shown in Figure 8 in Appendix F.2.1.

We experiment with both variants for Qwen-2.5-VL-Instruct-3B and 7B, training them with the same configurations as the original text SFT baseline. As shown in Table 7, the SFT variants with either coordinates or ASCII do not provide consistent significant improvements over the direct SFT baseline. Specifically, these variants with additional structural descriptions in either coordinates or ASCII yield slight gains in EM, but exhibit lower PR compared to the direct SFT baseline. Moreover, both variants still fall short of VPRL, suggesting that enriching textual input alone is insufficient to bridge the gap between visual and text-based planning.

**RL-trained text baseline.** We also explore the feasibility of applying RL to improve the planning performance with multimodal input, given the success of RL in the pure language planning domain (Guo et al., 2025a). We train an RL-based text model using Qwen-2.5-VL-Instruct-3B and 7B with the GRPO algorithm, with output format shown in Figure 8.

We adopt the same progress-based reward design as in VPRL for fair comparison, in addition to a simple *format reward* that ensures reasoning is enclosed within <think> tag and the final answer within <answer> tag.

- If the action is optimal (i.e., aligned with some optimal trajectory from the current state), it receives a reward of $+1$.
- If the action is valid but non-optimal, it receives $0$.
- If the action is invalid (e.g., colliding with walls or stepping into holes), it receives $-5$, and all subsequent actions are also treated as invalid.

To prevent reward hacking through excessively short rollouts, any sequence shorter than the length of the optimal trajectory is padded with invalid actions for the remaining steps, which are considered "stay-in-place" moves, in other words, invalid transitions. Finally, to make rewards comparable across sequences of varying lengths, we normalize the total reward by sequence length $\sum_{t=1}^{T} r_t/T$.

We train the text RL baselines for 10 epochs, consistent with VPRL. Using the VPRL progress reward described above, the model achieves 54.4% EM (Table 7). We suspect that this limited performance

---

**Input Example**

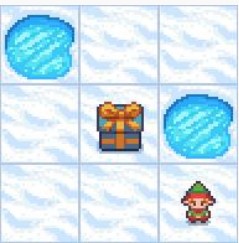

**Task:** Generate the shortest valid sequence of actions that moves the elf from the starting position to the goal without stepping into any ice holes (see detailed prompts in Appendix G).

---

**SFT (w/ Coordinates)**

After analysis, the layout of the given image is as follows:
- Starting position: (3, 3)
- Goal: (2, 2)
- Ice holes: (1, 1), (2, 3)
- Passable cells: (1, 2), (1, 3), (2, 1), (3, 1), (3, 2)

Based on the above layout, the final answer is: `<ANSWER>left up</ANSWER>`

---

**SFT (w/ ASCII)**

The symbols used to represent the grid are:
- S denotes the starting position, G the goal, H the ice hole, F passable cells.

HFH
FGH
FFS

Based on the above layout, the final answer is: `<ANSWER>left up</ANSWER>`

---

**GRPO**

```
<think>
```
1. The elf is in the bottom right cell.
2. The goal is in the top center cell.
3. The top left cell has an ice hole, so the elf cannot go left there.
4. The elf needs to move one cell left to the center cell.
5. Then the elf needs to move one cell up to the goal cell.

The sequence of actions are:
1. Move one cell left to the center cell.
2. Move one cell up to the goal cell.
```
</think>

<answer>left up</answer>
```

Figure 8: Examples of model outputs under different trained textual variants: SFT (w/ Coordinate), SFT (w/ ASCII), and GRPO. Each variant follows a distinct format for representing the environment. Even for the relatively simple 3 × 3 example input, we observe that the ASCII and GRPO models still produce erroneous layout descriptions.

is due to the reward design not being sufficiently discriminative. In particular, trajectories that contain the same number of optimal and non-optimal actions receive identical rewards regardless of their order (e.g., starting with optimal actions and then switching to non-optimal ones yields the same return as the reverse). As a result, the model tends to first learn to produce valid actions in general, and only later to distinguish optimal actions among them.

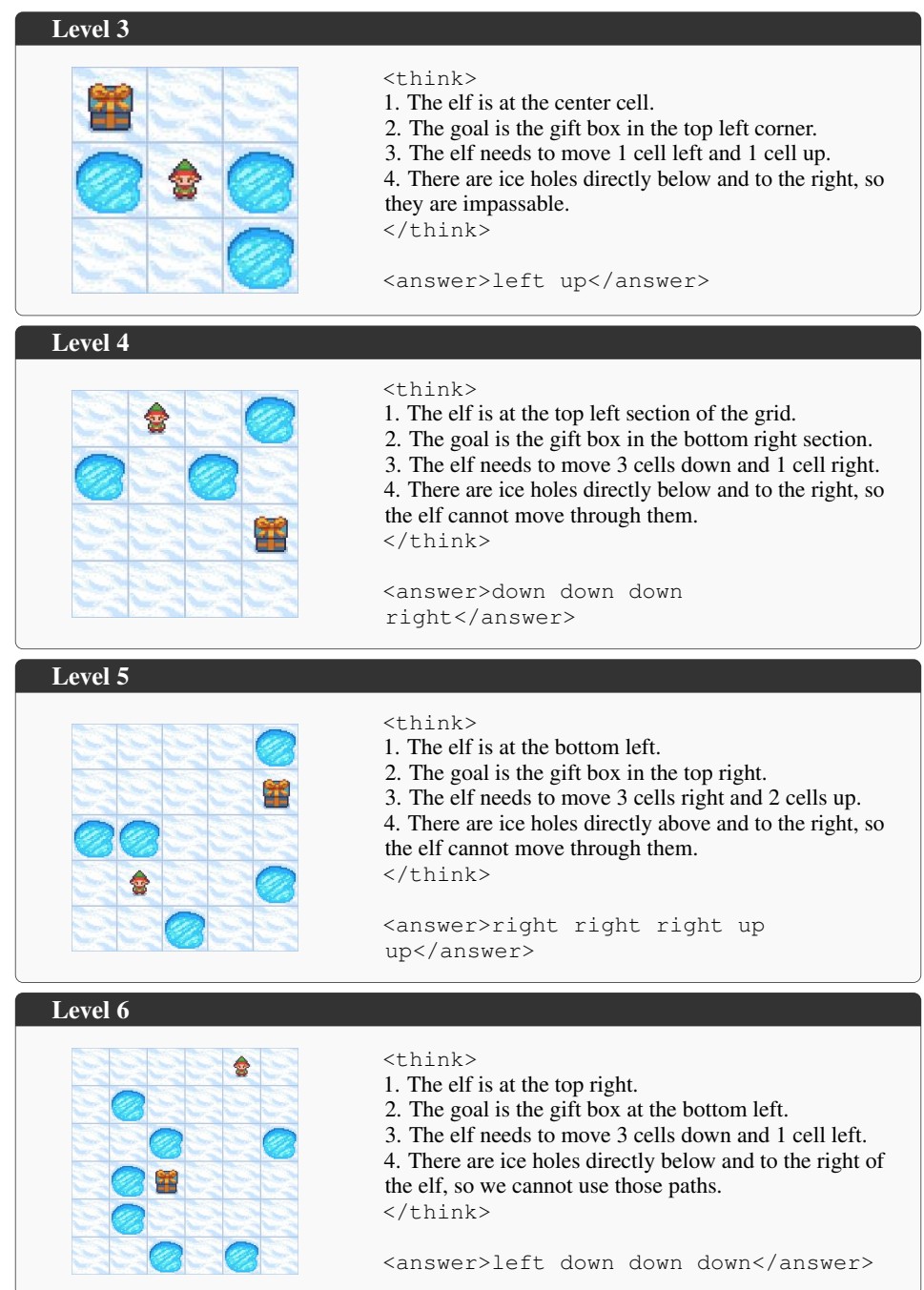

Figure 9: Examples of responses from the textual GRPO baseline with PR metric as the reward on FROZENLAKE across different difficulty levels. Each box shows the input image and the corresponding model output. In all cases, the model produces incorrect layout descriptions, which in turn lead to incorrect predicted action sequences.

To address this issue, we further design an alternative reward function by directly adopting the Progress Rate (PR) metric from the main paper. This formulation encourages the model to focus on generating consecutive valid forward moves from the start, rather than separating the learning of validity and optimality. Under the same training conditions, this reward improves EM to 60.1%, but the performance still lags behind the direct SFT baseline. As we discussed in Section 4 (error analysis paragraph), we attribute the bottleneck of language-based planning with RL to the modality

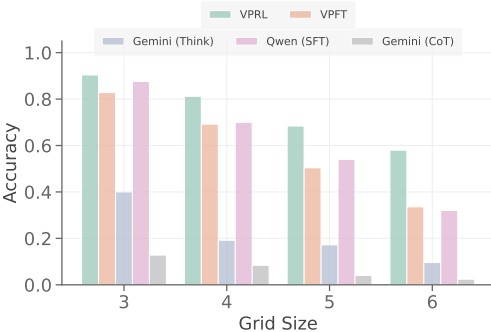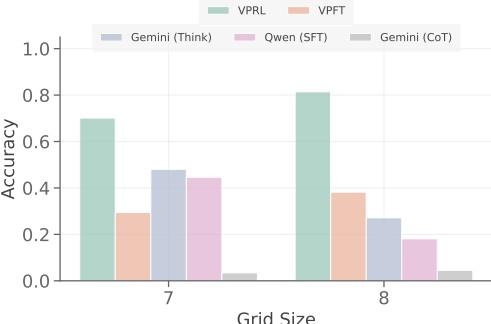

Figure 10: Performance across different grid sizes, reflecting task difficulty. **Left**: MAZE. **Right**: MINIBEHAVIOR. Visual planners consistently maintain higher accuracy and exhibit flatter performance curves, indicating robustness to increasing complexity.

gap, which introduces inaccuracies in grounding visual information into text, causing exploration to proceed from misinterpreted states and thereby reducing the overall effectiveness of learning. By contrast, our visual planning paradigm avoids this modality gap by operating directly in the visual domain, ensuring exploration within the correct state space via RL.

### F.2.1 EXAMPLES OF TRAINED TEXTUAL VARIANTS

Outputs of different textual variants are illustrated in Figure 8, including SFT with coordinate and ASCII representations, as well as GRPO with reasoning traces. Even for the relatively simple $3 \times 3$ input, and despite all variants producing the correct final predictions shown in the figure, we observe that the ASCII and GRPO models still generate erroneous layout descriptions: in the ASCII case, the passable cell at the top right is misclassified as an ice hole, while in the GRPO case, the goal position is incorrectly identified.

We also conduct further qualitative analysis of responses from the textual RL baseline trained with the PR metric as the reward (Figure 9). In all cases, the model produces incorrect layout descriptions, which in turn lead to incorrect predicted action sequences, highlighting the modality gap in grounding visual information into text.

### F.3 PERFORMANCE WITH SCALING DIFFICULTIES

We evaluate the performance of different methods with respect to task difficulty in MINIBEHAVIOR and MAZE, as shown in Figure 10. Our visual planners consistently achieve higher accuracy across all grid sizes and exhibit notably flatter performance curves, indicating greater robustness to increasing environment complexity.

Interestingly, in MINIBEHAVIOR, we observe that the accuracy of visual planners increases with grid size, which is in contrast to the trend exhibited by textual planners. We hypothesize that this is due to the fixed layout components in this task, specifically, the presence of only a table and a printer. This maintains consistent layout complexity across different grid sizes and allows knowledge acquired in smaller grids to generalize effectively to larger grids. This suggests that visual planning better captures and transfers structural patterns in the environment.

### F.4 OUT-OF-DISTRIBUTION PERFORMANCE

Figure 11 illustrates generated images from VPFT and VPRL on OOD scenarios across MAZE, FROZENLAKE, and MINIBEHAVIOR tasks. Notably, both models exhibit a certain level of visual generalization to unseen configurations, such as larger grids with finer step granularity, despite not encountering them during training.

We subsequently quantitatively test generalization by evaluating the model on OOD environments with larger grid sizes. We find that SFT models perform poorly, while VPRL still demonstrates a

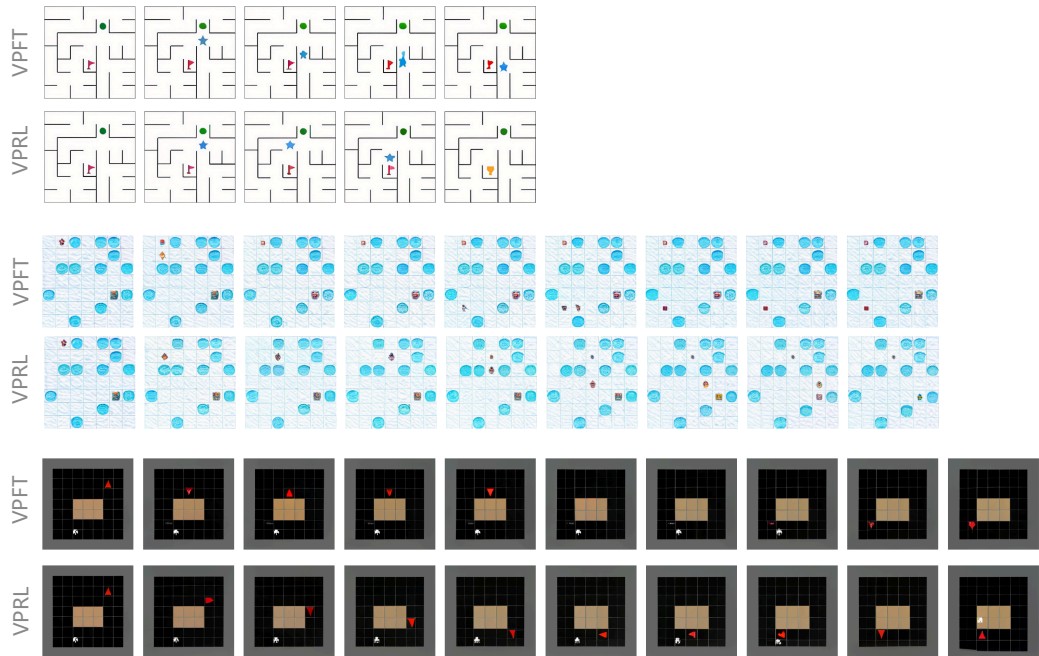

Figure 11: Qualitative comparison of visual planning outputs from VPFT (top) and VPRL (bottom) on out-of-distribution (OOD) scenarios with unseen larger grid size across MAZE, FROZENLAKE, and MINIBEHAVIOR. Each example shows a failure case from VPFT contrasted with a successful trajectory generated by VPRL under the same environment configuration.

certain level of visual planning capability, as shown in Table 9. VPRL consistently outperforms VPFT in both Exact Match and Progress Rate, suggesting that it, to some degree, captures underlying planning strategies rather than merely memorizing training patterns.

Finally, we analyze the robustness of VPRL by qualitatively testing its behavior under perturbed inputs. As shown in Figure 12, we mask portions of the input images with black or gray patches to simulate partial occlusion of the environment. Remarkably, the model continues to produce coherent planning traces within the masked environments, while preserving structural consistency with the visible input regions. This observation highlights the generalization capability of our visual planner, as it adapts to incomplete visual information without deviating from the underlying environment layout.

## F.5  ABLATION: THE ROLE OF STAGE 1

To better understand the role of Stage 1 in our two-stage framework, we conduct an ablation study isolating its impact. The primary purpose of Stage 1 is not to improve planning performance directly, but rather to initialize a policy with strong exploration capacity and valid output formats. To verify this, we reuse the original VPFT training pipeline, i.e., learning from optimal trajectories, but start from the Stage 1 checkpoint as VPFT*. Surprisingly, this variant yields lower final performance on FROZENLAKE compared to standard VPFT. This result supports our hypothesis that Stage 1 does not contribute to planning ability itself, but instead provides an exploration-friendly initialization that facilitates effective reinforcement learning in Stage 2.

## F.6  VISUAL PLANNING RESULTS

**VPRL Stage 1 and Stage 2.** Table 10 presents results for each stage of VPRL. After Stage 1, the model learns to generate plausible images but lacks goal-directed behavior, resulting in near-random performance across tasks. In Stage 2, reinforcement learning instills purposeful planning, enabling the model to align generations with the goal and outperform VPFT across all benchmarks.

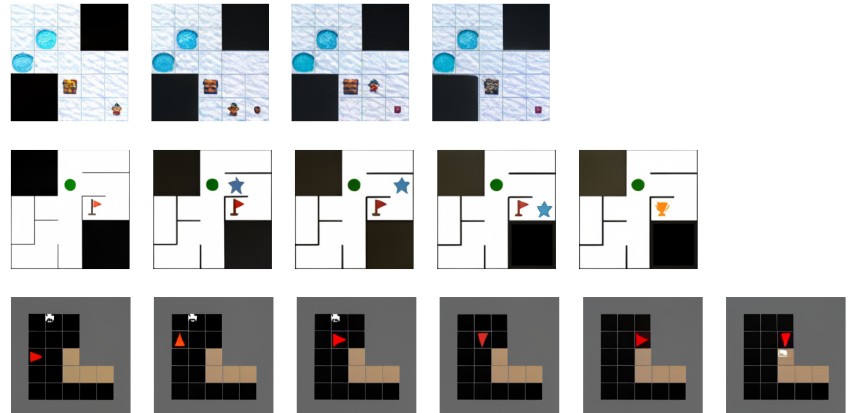

Figure 12: Qualitative analysis of VPRL under perturbed inputs (the first image of each trace). When parts of the input environment are masked (black/gray regions), VPRL maintains consistent planning traces aligned with the visible structure, demonstrating robustness to incomplete visual information without deviating from the underlying environment layout.

Table 10: Performance comparison of VPRL Stage 1 and Stage 2 across all three tasks.

| Model | FROZENLAKE | | MAZE | | MINIBEHAVIOR | |
|---|---|---|---|---|---|---|
| | EM (%) | PR (%) | EM (%) | PR (%) | EM (%) | PR (%) |
| VPRL Stage 1 | 11.1 | 27.2 | 9.6 | 22.7 | 0.5 | 14.2 |
| **VPRL Stage 2** | **91.6** | **93.2** | **74.5** | **77.6** | **75.8** | **83.8** |

**Generated Visual Planning Traces for Illustration.** Figure 13 shows the generated visual planning traces for FROZENLAKE, with Figure 14 for MAZE and Figure 15 for MINIBEHAVIOR. Each visual trajectory begins with the initial state as the input (the first frame), followed by a sequence of intermediate states generated by VPRL that form the predicted visual plan.

We include examples from three categories: (1) **Optimal cases**, where the model successfully generates the shortest valid path to the goal; (2) **Non-optimal cases**, where the agent fails to reach the goal within the optimal number of steps due to intermediate non-optimal actions; and (3) **Invalid cases**, in which the generated trajectory contains invalid actions that violate environment constraints, preventing task completion. Notably, as illustrated in Figure 3, we still observe occasional planning errors. While reinforcement learning significantly improves generalization compared to supervised fine-tuning, it does not fully eliminate such failure cases.

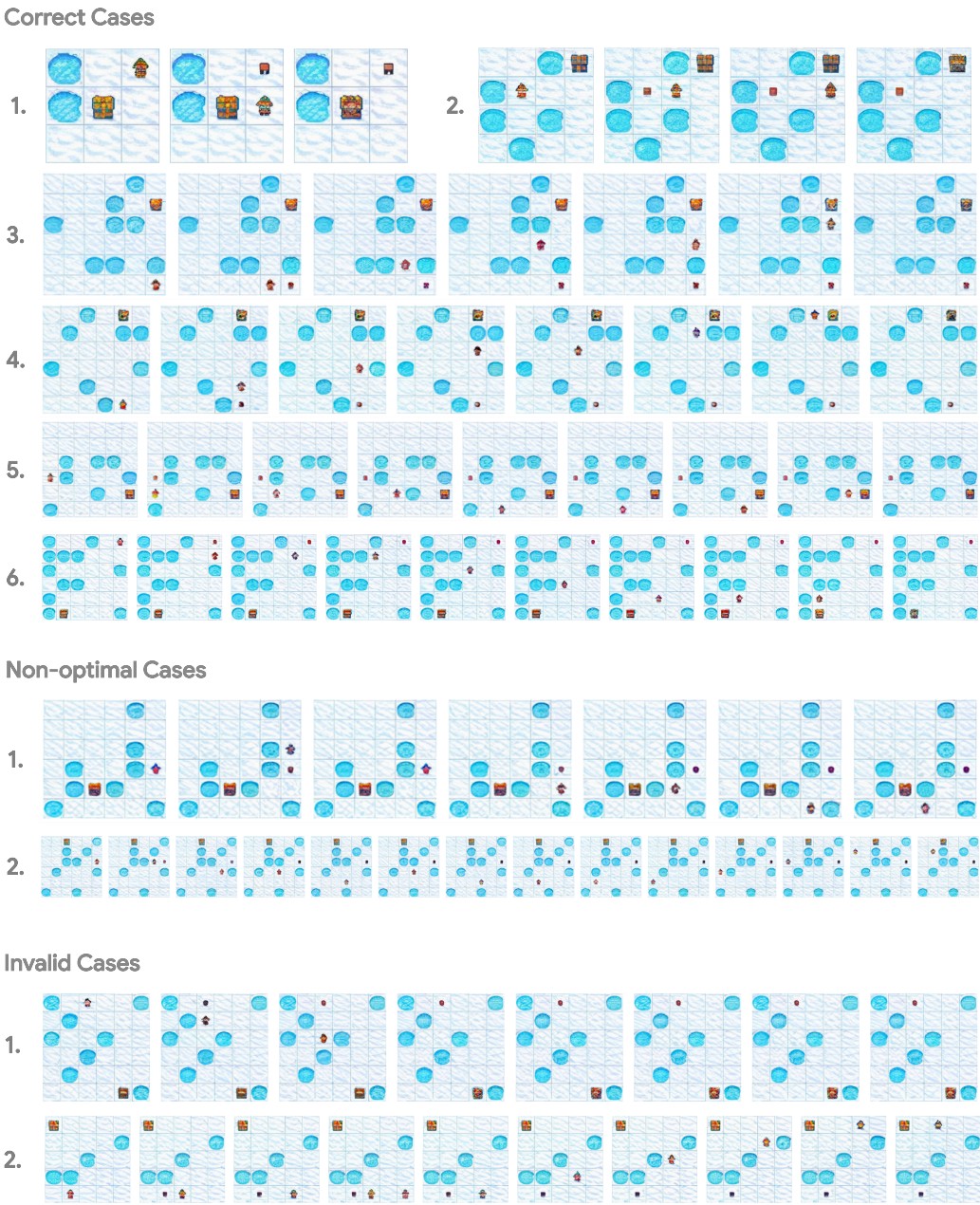

Figure 13: Generated visual planning trajectories from VPRL on the FROZENLAKE test set. We illustrate three representative categories: optimal, non-optimal, and invalid cases. In non-optimal examples, the model occasionally enters local loops but still has the chance to make progress toward the goal, see the first and third trajectories. In invalid cases, despite a significant reduction in failure rate, VPRL still exhibits errors such as disappearing agents, contradictory actions (e.g., simultaneous left and right), or unrealistic teleportation.

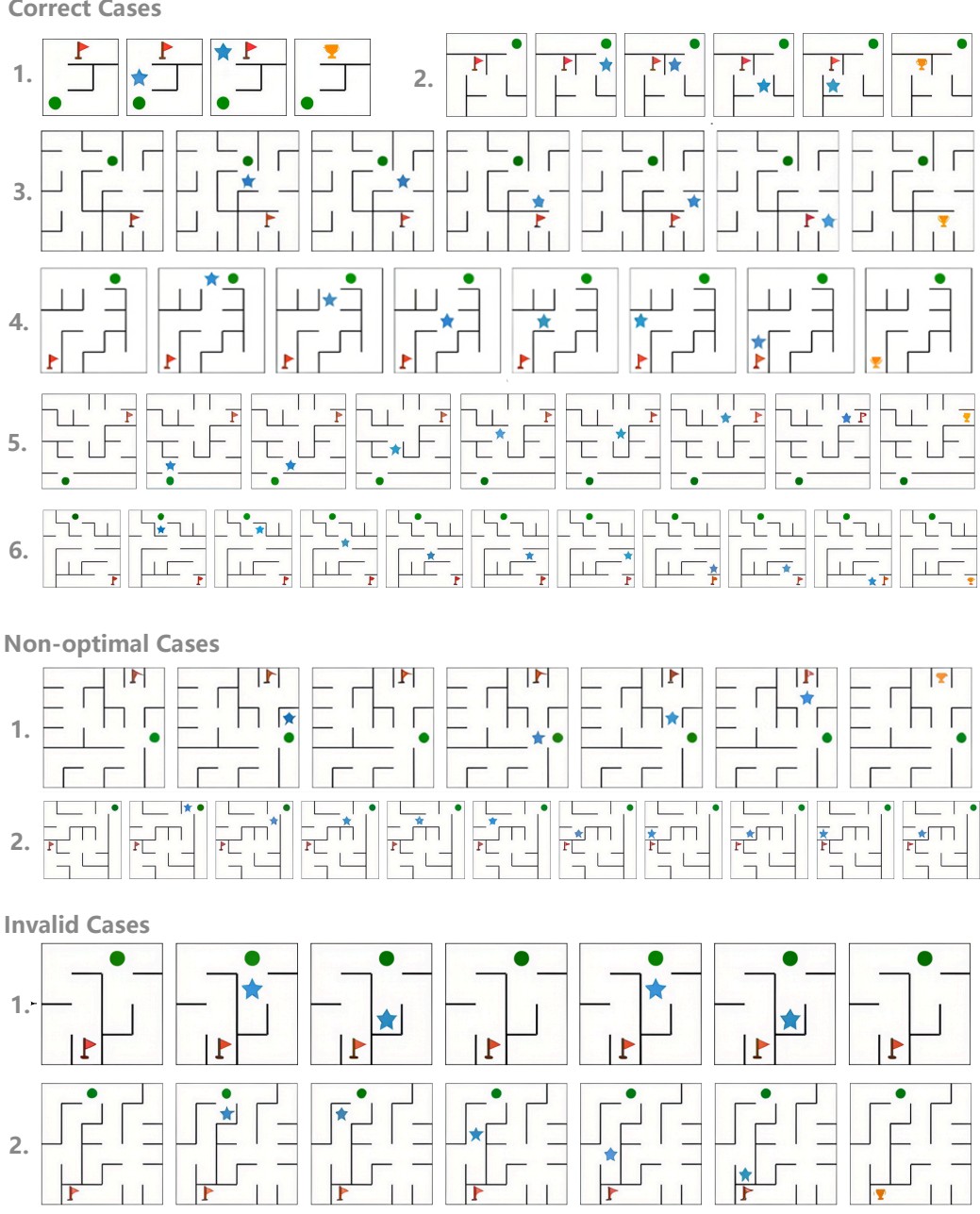

Figure 14: Generated visual planning trajectories from VPRL on the MAZE test set. We illustrate three representative categories: optimal, non-optimal, and invalid cases. In non-optimal examples, similar to FROZENLAKE, the model occasionally enters redundant loops but still progresses toward the goal. Invalid cases include maze-specific errors, such as the agent erroneously traversing through walls, violating the structural constraints of the environment. Notably, we observe that in the last invalid case, the agent is able to plan an optimal trajectory in subsequent steps.

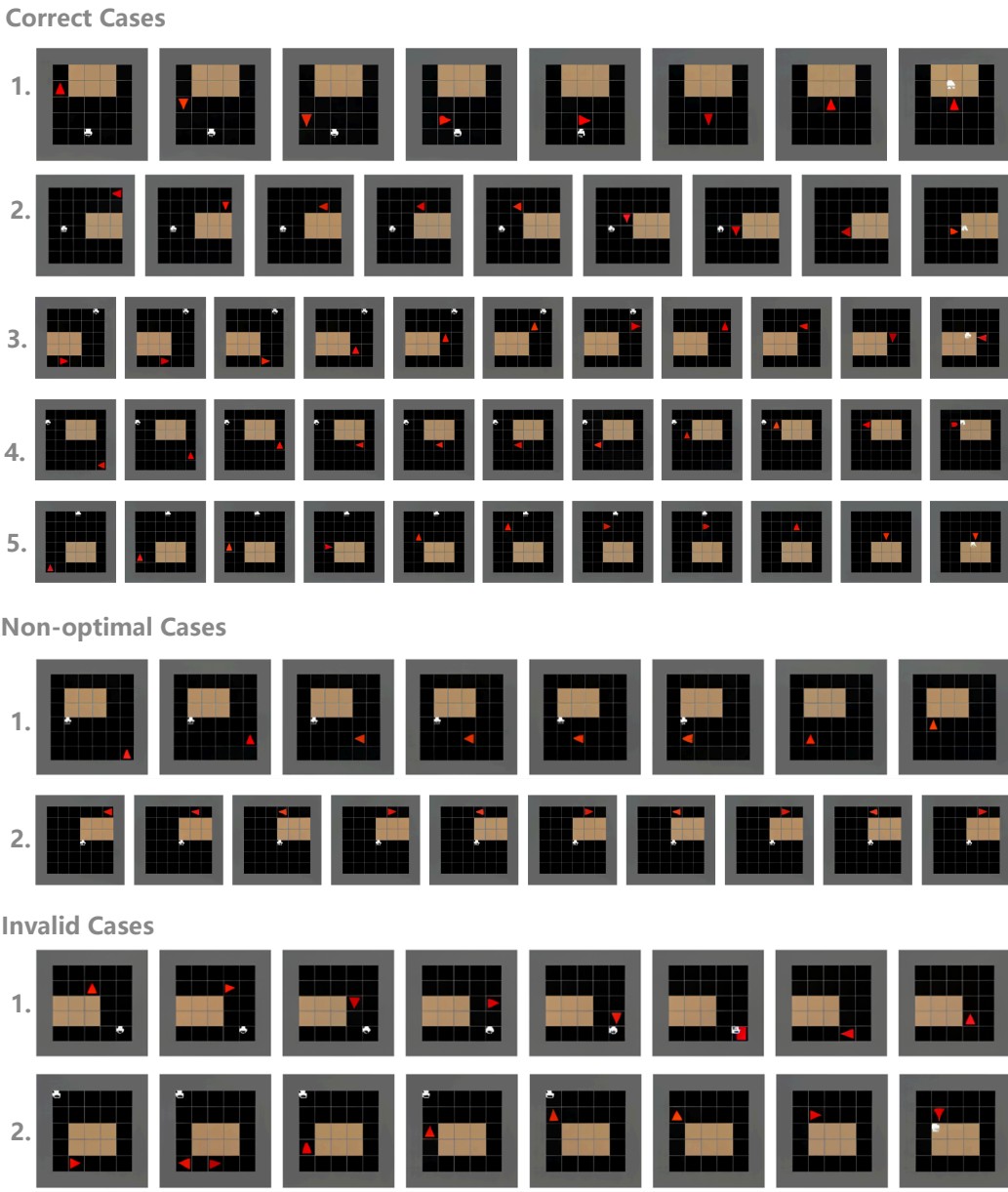

Figure 15: Generated visual planning trajectories from VPRL on the MINIBEHAVIOR test set.

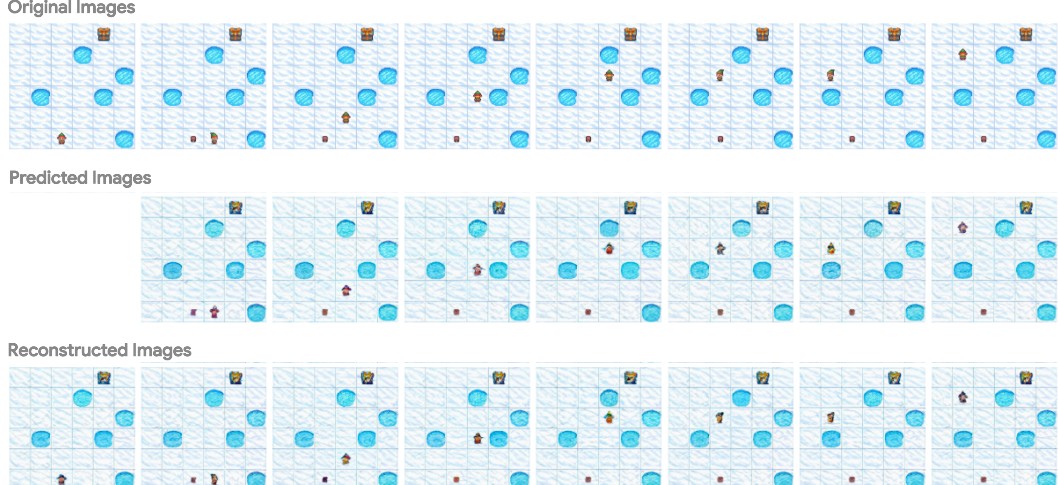

Figure 16: Qualitative comparison between original images (top), predicted images by the model (middle), and reconstructed images obtained by encoding and decoding the original inputs (bottom).

Table 11: Exact Match (EM) and Progress Rate (PR) on FROZENLAKE under VPRL when using ground-truth images versus self-generated images as inputs during inference.

| Model | EM (%) | | | | | PR (%) | | | | |
|---|---|---|---|---|---|---|---|---|---|---|
| | L3 | L4 | L5 | L6 | Avg. | L3 | L4 | L5 | L6 | Avg. |
| VPRL | | | | | | | | | | |
| - (w/ self-generated images) | 97.6 | **95.6** | 90.8 | **82.4** | 91.6 | 98.4 | **96.0** | 93.0 | **85.6** | 93.2 |
| - (w/ ground-truth images) | **98.4** | 95.2 | **93.2** | 81.6 | **92.1** | **98.5** | 95.8 | **94.1** | 85.3 | **93.4** |

## F.7    IMAGE QUALITY ANALYSIS

It can be observed that the intermediate images on FROZENLAKE generated by the visual planner in Figure 4 contain noticeable artifacts, and we suspect that this noise arises from the limitation of the image tokenizer rather than from the model's image generation ability. To verify this, we include an additional analysis on FROZENLAKE that illustrates how the tokenizer reconstructs images in our framework.

**Limitations of the Image Tokenizer.** Figure 16 confirms that the artifacts observed in our predicted images originate from the tokenizer rather than from the prediction process itself. When encoding a ground-truth image into visual tokens and decoding it back, the reconstructed output shows similar artifacts inevitably introduced by the tokenizer to those in the model's predictions, which makes the reconstruction not identical to the original image. At the same time, we observe that the intermediate images produced by the model are already comparable in quality to the reconstructed images. While our work focuses on planning rather than image generation quality, this observation indicates that the visual planner generates images that are sufficient for effective planning.

We consider this behavior to be encouraged by the dynamics interpreter. During the training, the dynamics interpreter serves as an implicit format constraint. Any generated image that it cannot parse is treated as an invalid transition and receives a penalty, enforcing the model to maintain the semantic structure of the environment in its generated images.

**Robustness to Intermediate Image.** We subsequently conduct a quantitative study to evaluate whether providing high-quality intermediate images at inference improves performance. Instead of feeding back the model's self-generated image at each step, we replace it with the ground-truth image rendered by the environment, which serves as a high-quality version.

Table 12: Average inference token cost across FROZENLAKE, MAZE, and MINIBEHAVIOR. We also report the average of the task-level average costs. Higher values indicate higher computational cost.

| Model | FROZENLAKE | MAZE | MINIBEHAVIOR | Avg. |
|---|---|---|---|---|
| Closed-Source Models | | | | |
| Gemini 2.0 Flash | | | | |
|   - Direct | 10.8 | 12.5 | 14.8 | 12.7 |
|   - CoT | 150.5 | 166.5 | 196.5 | 171.2 |
| Gemini 2.5 Pro (*think*) | **885.6** | **1030.2** | **1619.9** | **1178.6** |
| Open-Source Models | | | | |
| Qwen 2.5-VL-Instruct-7B | | | | |
|   - Direct | 13.4 | 95.9 | 13.9 | 41.1 |
|   - CoT | 306.2 | 316.4 | 272.3 | 298.3 |
|   - SFT | 10.7 | 11.4 | 13.2 | 11.8 |
| LVM-7B | | | | |
|   - VPFT (ours) | 819.2 | 957.2 | 1471.2 | 1082.5 |
|   - VPRL (ours) | 819.2 | 957.2 | 1471.2 | 1082.5 |

Table 11 shows that the performance with and without high-quality images remains similar across all grid sizes. This shows that our visual planner is robust to visual noise and does not depend on perfectly rendered images to plan effectively.

## F.8 COMPUTATIONAL COST ANALYSIS

To provide a quantitative comparison of the computational cost between visual planning and traditional textual reasoning, we further analyse the token usage of both the visual planner and the textual baselines during inference. We compute the average number of generated tokens for all models reported in Table 1 across all tasks. In addition, we include a more detailed breakdown of the token cost for the trained textual planner variants listed in Table 2, evaluated on FROZENLAKE.

Table 12 and Table 13 summarise the resulting inference token cost. As expected, visual planning introduces a noticeable computational overhead due to repeated image generation. However, this additional cost remains afford-

Table 13: Average inference token cost of trained textual planner variants on FROZENLAKE.

| Model | Token Cost |
|---|---|
| Qwen 2.5-VL-Instruct-7B | |
| - SFT | |
|   - *Direct* | 10.7 |
|   - *w/ Coordinates* | 179.0 |
|   - *w/ ASCII* | 84.3 |
| - GRPO | |
|   - *w/ VPRL progress reward* | 129.8 |
|   - *w/ PR metric reward* | 74.9 |

able in practice when compared with textual CoT. On average across the three tasks, the token cost of our visual planner is roughly 3 times that of Qwen 2.5-VL-Instruct-7B with CoT and around 6 times that of Gemini 2.0 Flash with CoT, suggesting that our method is still computationally feasible. We also observe that thinking models, such as Gemini 2.5 Pro, produce the largest number of tokens among all tasks, indicating that visual planning is not always the most expensive option.

## G   PROMPTING TEMPLATES

**FROZENLAKE (Direct)**

```
Task: Frozen Lake Shortest Path Planning

You are given an image of a grid-based environment. In this environment:
- An elf marks the starting position.
- A gift represents the goal.
- Some cells contain ice holes that are impassable for the elf.
- The elf can move in one of four directions only: "up", "down", "left",
    or "right". Each move transitions the elf by one cell in the
    corresponding absolute direction. Diagonal movement is not permitted.

Your task is to analyze the image and generate the shortest valid
    sequence of actions that moves the elf from the starting position to
    the goal without stepping into any ice holes.

Provide your final answer enclosed between <ANSWER> and </ANSWER>, for
    example: <ANSWER>right up up</ANSWER>.
```

**FROZENLAKE (Coordinate & ASCII Representation)**

```
Task: Frozen Lake Shortest Path Planning

You are given an image of a grid-based environment. In this environment:
- An elf marks the starting position.
- A gift represents the goal.
- Some cells contain ice holes that are impassable for the elf.
- The elf can move in one of four directions only: "up", "down", "left",
    or "right". Each move transitions the elf by one cell in the
    corresponding absolute direction. Diagonal movement is not permitted.

Your task is to analyze the image and generate the shortest valid
    sequence of actions that moves the elf from the starting position to
    the goal without stepping into any ice holes.

Describe the layout of the environment based on your analysis of the
    image, then provide your final answer enclosed between <ANSWER> and
    </ANSWER>, for example: <ANSWER>right up up</ANSWER>.
```

**FROZENLAKE (GRPO)**

```
Task: Frozen Lake Shortest Path Planning

You are given an image of a grid-based environment. In this environment:
- An elf marks the starting position.
- A gift represents the goal.
- Some cells contain ice holes that are impassable for the elf.
- The elf can move in one of four directions only: "up", "down", "left",
    or "right". Each move transitions the elf by one cell in the
    corresponding absolute direction. Diagonal movement is not permitted.

Your task is to analyze the image and generate the shortest valid
    sequence of actions that moves the elf from the starting position to
    the goal without stepping into any ice holes.

Present your reasoning enclosed within <think> and </think> tags. For
    example:
<think>Reasoning steps go here.</think>

Then, provide your final answer enclosed within <answer> and </answer>
    tags. For example:
<answer>right up up</answer>
```

## MAZE

```
Task: Maze Shortest Path Planning

You are given an image of a maze environment. In this environment:
- A green circle marks the starting position of the agent.
- A red flag marks the goal.
- The agent can move in one of four cardinal directions only: "up", "down
    ", "left", or "right". Each move shifts the agent by exactly one cell
     in that direction. Diagonal movement is not permitted.
- The black maze walls are impassable. The agent cannot pass through any
    wall segment.

Your task is to analyse the image and produce the shortest valid sequence
    of actions that moves the agent from its starting position to the
    goal without crossing any wall.

Provide your final answer enclosed between <ANSWER> and </ANSWER>, for
    example: <ANSWER>right up up</ANSWER>.
```

## MINIBEHAVIOR

```
Task: Mini-Behavior Installing the Printer

You are given an image of a grid-based environment. In this environment:
- The red triangle represents the agent.
- The white icon represents the printer, which must be picked up by the
    agent.
- The brown tiles represent the table, where the printer must be placed.

The agent can take the following actions:
- "up", "down", "left", "right": each action shifts the agent by exactly
    one cell in that direction. Diagonal movement is not permitted.
- "pick": pick up the printer if it is in one of the four adjacent cells
    surrounding the agent. This action is invalid if there is no adjacent
     printer.
- "drop": drop the printer onto the table if the agent is adjacent to a
    table cell. This action is invalid if there is no adjacent table.

Constraints:
- The agent cannot move through the table tiles.
- The agent cannot move through the printer until it has been picked up.
    After picking it up, the agent may move through the cell that
    previously contained the printer.

Your task is to analyse the image and produce the shortest valid sequence
    of actions that allows the agent to pick up the printer and then
    place it on the table.

Provide your final answer enclosed between <ANSWER> and </ANSWER>, for
    example: <ANSWER>right down right pick left drop</ANSWER>.
```

