# OpenReview forum: "Visual Planning: Let's Think Only with Images"
_ICLR.cc/2026/Conference — ICLR 2026 Oral_

### Official Review · Reviewer_Rhdx · 2025-10-31

**Soundness:** 4
**Presentation:** 4
**Contribution:** 3
**Rating:** 8
**Confidence:** 4

**Summary:**

This study proposes a visual planning approach that performs task planning entirely through visual representations. In this paradigm, planning is carried out via a sequence of images, which encode the step-by-step reasoning process in the visual domain, similar to how humans make sketches or visualize future actions.

**Strengths:**

It performs planning purely within the visual modality as a holistic process, where the actions are not explicitly predicted but instead implicitly represented by transitions between visual states.

**Weaknesses:**

The experimental validation is restricted to discrete, low-dimensional grid-world navigation tasks, where the visual states are comparatively simple and straightforward to encode.
Visual planning implicitly expresses actions by generating a sequence of visual states. Although this avoids modality switching, when planning fails, the absence of an explicit action sequence (such as textual CoT) makes the model’s decision process difficult to debug and understand.

**Questions:**

1. Visual planning involves generating high-dimensional image sequences, which can be computationally more expensive than searching in low-dimensional text/action spaces as in language models. Please elaborate on how  it's computational efficiency and search space complexity compare to textual CoT methods?

---

> ### Author Response · Authors · 2025-11-24
> **Response to Reviewer Rhdx by Authors (Part 1)**
>
> We thank the reviewer for the positive and insightful comments. We also appreciate the reviewer raising concerns about environment complexity, interpretability, and computational cost. We address each point below.
>
> > The experimental validation is restricted to discrete, low-dimensional grid-world navigation tasks, where the visual states are comparatively simple and straightforward to encode.
>
> While we agree that our current experiments focus on 2D grid-based tasks, our goal in this paper is to validate the feasibility of a new visual-only paradigm. Grid-based environments provide a controlled setup where states and transitions are fully observable, which allows us to test this idea without introducing additional noise.
>
> Within this setup, we believe our tasks still exhibit meaningful variation in complexity. FrozenLake contains the richest environment patterns, and MiniBehavior introduces two additional action types ("pick" and "drop"). These differences allow us to examine whether the proposed framework remains effective across different 2D environments, illustrating that our paradigm has the potential to be extended to more challenging settings such as 3D environments. Investigating and validating this extension is a natural next step, and we have highlighted this direction in the future work of our revised manuscript (L849-857).
>
> > Visual planning implicitly expresses actions by generating a sequence of visual states. Although this avoids modality switching, when planning fails, the absence of an explicit action sequence (such as textual CoT) makes the model’s decision process difficult to debug and understand.
>
> This connects directly to the central aim of our work. One core motivation of the paper is to test whether a model can plan through visual representations without relying on textual rationales or symbolic intermediate representations.
>
> While the model does not output textual CoT action sequences, the predicted images provide a clear visual trace of the model's decisions. By inspecting these visual rationale step by step, we can identify where the model deviates from valid transitions or begins to accumulate errors, which still brings the interpretability.

---

> > ### Author Response · Authors · 2025-11-24
> > **Response to Reviewer Rhdx by Authors (Part 2)**
> >
> > > Visual planning involves generating high-dimensional image sequences, which can be computationally more expensive than searching in low-dimensional text/action spaces as in language models. Please elaborate on how it's computational efficiency and search space complexity compare to textual CoT methods?
> >
> > We thank the reviewer for raising the question about computational cost and search complexity. To examine this, we report the average output token length for our visual planner and the textual baselines, which is added in Appendix F.8. The tables below summarise the results across all tasks for the reviewer's convenience:
> >
> > | Model               | FrozenLake | Maze | MiniBehavior |
> > |-------------------------------|-----------:|-----:|-------------:|
> > | Gemini 2.0 Flash - Direct | 10.8      | 12.5 | 14.8 |
> > | Gemini 2.0 Flash - CoT    | 150.5     | 166.5 | 196.5 |
> > | Gemini 2.5 Pro (think)         | **885.6**  | **1030.2** | **1619.9** |
> > | Qwen 2.5-VL-Instruct-7B -  Direct | 13.4 | 95.9 | 13.9 |
> > | Qwen 2.5-VL-Instruct-7B - CoT    | 306.2 | 316.4 | 272.3 |
> > | Qwen 2.5-VL-Instruct-7B - SFT (direct)   | 10.7 | 11.4 | 13.2 |
> > | VPFT (ours)             |  819.2         | 957.2     | 1471.2 |
> > | VPRL (ours)             |  819.2         | 957.2     | 1471.2 |
> >
> > We also include the cost of the trained textual planner variants on FrozenLake:
> >
> > | Model                    | FrozenLake |
> > |----------------------------------|---------------------|
> > | Qwen 2.5-VL-Instruct-7B - SFT (direct)                       | 10.7               |
> > | Qwen 2.5-VL-Instruct-7B - SFT (w/ Coordinates)               | **179.0**              |
> > | Qwen 2.5-VL-Instruct-7B - SFT (w/ ASCII)                     | 84.3               |
> > | Qwen 2.5-VL-Instruct-7B - GRPO (w/ VPRL progress reward)      | 129.8              |
> > | Qwen 2.5-VL-Instruct-7B - GRPO (w/ PR metric reward)          | 74.9               |
> >
> > We agree that visual outputs introduce additional computational overhead relative to symbolic reasoning, which is confirmed by the table. We would also like to emphasize that this cost reflects a tradeoff for the better performance visual planning achieves in visual-centric tasks, where the textual reasoning paradigm falls short, even with fewer tokens. We also observe that thinking models, such as Gemini 2.5 Pro, produce the largest number of tokens among all tasks, indicating that visual planning is not always the most expensive option.
> >
> > Regarding search complexity, our visual planner operates within a fixed codebook of 8192 visual tokens. It generates each image by selecting 256 tokens from this set, which results in a stable and bounded search space. In contrast, textual models generate tokens from a much larger vocabulary. For example, Qwen 2.5-VL uses a vocabulary of ~150k tokens, leading to a much larger space of possible outputs when producing a CoT rational.

---

### Official Review · Reviewer_uu3P · 2025-10-31

**Soundness:** 3
**Presentation:** 4
**Contribution:** 3
**Rating:** 6
**Confidence:** 2

**Summary:**

The paper takes visuals as a medium for expressing and structuring reasoning, which potentially learn spatial and geometrical information. To realize this, the author proposes Visual Planning, planning through purely visual representations for “vision-first” tasks. The proposed GRPO demonstrates improvement in planning.

**Strengths:**

The author investigates the potential of visual representation as a medium, which expands the research of LLMs to a broader area.

The presentation of the paper is great, with a clear statement and an appropriate graph.

The paper is the first attempt to investigate whether models can achieve planning purely through visual representations.

**Weaknesses:**

Can you provide any figures to clearly show the difference between language as a medium and visual as a medium in certain cases?

It will be better if we can discuss any advantages of visual as a medium in real CV tasks, such as visual grounding. And the proposed methods, whether they can be easily transferred to 3D?

If we finally want to get an MLLM, how do we add GRPO to the regular training receipt? When to align visuals with other modalities?

**Questions:**

See Weakness.

---

> ### Author Response · Authors · 2025-11-24
> **Response to Reviewer uu3P by Authors (Part 1)**
>
> We appreciate the reviewer's valuable comments and recognition of our work. Please see our detailed response to the concerns on comparison clarity, transferability, and potential combination with MLLM.
>
> > Can you provide any figures to clearly show the difference between language as a medium and visual as a medium in certain cases?
>
> We kindly refer the reviewer to Figure 4, where Gemini 2.5 Pro incorrectly describes a 5×5 grid as 5×7 and misidentifies the positions of ice holes in the text. Similar errors can also be observed in our post-trained textual baselines (Figures 8 and 9 in the Appendix), where textual descriptions often fail to reproduce the correct layouts. These examples highlight a clear limitation of relying on language as an intermediate medium. In contrast, our method avoids this modality gap and ensures exploration within the correct state space by reasoning directly in the visual domain.
>
>  > It will be better if we can discuss any advantages of visual as a medium in real CV tasks, such as visual grounding.
>
> We believe visual planning aligns naturally with many real-world CV tasks, as visual signals could retain structural relationships, such as spatial layout, object interactions, and geometric constraints, which are often more difficult to encode in text. Most CV Tasks rely heavily on such spatial structure, suggesting that planning directly over visual states can be advantageous.
>
>  > And the proposed methods, whether they can be easily transferred to 3D?
>
> In our current settings, we choose grid-based environments for controllability, spanning different levels of pattern complexity (from simple Maze to visually richer FrozenLake) and action complexity (Maze and FrozenLake with 4 different actions and MiniBehavior with 6 actions). Experimental results show that our method performs well across these levels of complexity, illustrating its potential to be extended to more challenging environments. In this case, we believe our core reward design is broadly applicable to 3D planning tasks, as actions in most planning settings can naturally be categorized into one of three types (optimal, sub-optimal, and invalid). In our view, the main difficulty lies not in the reward formulation, but in defining reliable progress signals when visual transitions become more complex. This can be addressed by learning a reward model that evaluates state transitions directly from images, or by using trajectory-level rollouts, where the model uses final success signals to identify actions that lead to successful outcomes. We consider investigating this extension a natural next step and have highlighted this direction in the revised manuscript (L849-857).

---

> > ### Author Response · Authors · 2025-11-24
> > **Response to Reviewer uu3P by Authors (Part 2)**
> >
> > > If we finally want to get an MLLM, when to align visuals with other modalities?
> >
> > We thank the reviewer for raising this question, and we agree that this is an interesting topic for discussion. In our work, visual planning does not rely on strict alignment between visual and textual modalities. The planner operates in a purely visual space, where states and actions are represented and updated through visual signals. This design suggests a novel dual-channel reasoning structure in MLLM, in which visual and textual reasoning pathways can remain separate and interact only when needed. For example, when a task is visually friendly and can be solved entirely within the visual channel, the model can perform all internal reasoning visually and only align with the language channel at the final stage to describe the solution.
> >
> > > If we finally want to get an MLLM, how do we add GRPO to the regular training receipt?
> >
> > We have not explored this integration step in our current work, but we see it as a promising direction for future work and as a new avenue that could broaden the scope of multimodal research. Existing MLLM training pipelines rely on the textual tokens as the carrier of reasoning. Therefore, a possible way to integrate GRPO is to allow the MLLM to treat visual tokens as an alternative reasoning pathway during training. Instead of generating only text-based CoT, the MLLM could be trained to generate short visual rollouts as intermediate reasoning states, and our two-stage training pipeline would guide these visual transitions in the same way it guides VPRL. This does not require changing the core MLLM pipeline, but simply adds a new reasoning option that the model can choose to use when it is beneficial. In our view, the main challenge lies in allowing the model to learn when cross-modal interaction is helpful and when each channel can operate on its own. One potential approach is to design training signals that reward the model for choosing the modality that best reflects the underlying task structure, so that the two channels can coordinate without forcing unnecessary alignment. We advocate further investigation in this direction and believe that our visual planning paradigm can contribute to the development of a more unified and flexible multimodal reasoning framework.

---

### Official Review · Reviewer_2U49 · 2025-10-31

**Soundness:** 3
**Presentation:** 3
**Contribution:** 3
**Rating:** 4
**Confidence:** 4

**Summary:**

This paper introduces Visual Planning, a new paradigm that performs reasoning through visual representations rather than language. Using VPRL (Visual Planning via Reinforcement Learning), a GRPO-based post-training framework for large vision models, our approach generates image sequences that visualize step-by-step inference. Experiments on visual navigation tasks (FrozenLake, Maze, MiniBehavior) show that VPRL surpasses text-based reasoning, highlighting visual reasoning as a powerful complement to language reasoning for spatial and geometry-driven problems.

**Strengths:**

1. New Paradigm: The paper introduces "Visual Planning" as a genuinely new paradigm for reasoning.

2. Good Empirical Results: The proposed method, Visual Planning via Reinforcement Learning (VPRL), significantly outperforms a wide range of baselines.

3. Methodological Robustness: The two-stage VPRL framework is well-designed and justified.

**Weaknesses:**

1. Reliance on an External Oracle for Rewards: A significant weakness in the method's detail is its reliance on non-learned, external modules to provide the reward signal. The VPRL framework depends on a "dynamics interpreter" and a "progress estimator". The appendix reveals this estimator is a Breadth First Search (BFS) algorithm —an oracle that has already solved the task and knows the optimal path from every state. The interpreter also uses rule-based pixel and IoU comparisons. This means the model isn't learning the environment's dynamics or the concept of progress; it's learning to generate images that satisfy an external oracle that already has the answers.

2. Insufficient Justification for a "Purely Visual" Paradigm: This paper needs to justify why exploring a purely non-verbal, vision-only paradigm is a necessary or superior research direction. The authors' decision to "eliminate language as a confounding factor" is a research-scoping choice, but it is not a strong argument for the paradigm's utility.

3. Limited Task Complexity and Scalability: The paradigm is validated only on simple, 2D, discrete grid-world environments (FROZENLAKE, MAZE, MINIBEHAVIOR). It is highly questionable if this approach can scale to complex, 3D, photorealistic, or continuous-state environments (e.g., robotics). In such settings, autoregressively generating a perfect, step-by-step sequence of future images is computationally expensive (a point the authors concede ) and the rule-based reward function (which relies on pixel comparison ) would be far too brittle to work.

**Questions:**

All of my qeustions are listed in the weakness section. If my concerns are well addressed, I will raise my rating.

---

> ### Author Response · Authors · 2025-11-24
> **Response to Reviewer 2U49 by Authors (Part 1)**
>
> We thank the reviewer for their constructive comment and appreciate their positive remarks regarding the originality of our visual planning paradigm. Below, we address the concerns about reward design and computational cost. We also give a clearer explanation of the motivation for our work. We hope our response helps resolve the concerns and that the reviewer could consider improving the score.
>
> > Reliance on an External Oracle for Rewards: The model isn't learning the environment's dynamics or the concept of progress; it's learning to generate images that satisfy an external oracle that already has the answers.
>
> We acknowledge that our method relies on an oracle to compute the progress reward, which checks whether the agent makes progress toward the goal. However, we would like to clarify that this design is aligned with standard Reinforcement Learning with Verifiable Rewards (RLVR) practice [1] in LLM, where an external evaluator provides a verifiable reward by comparing the model's output with the correct solution, which also acts as an oracle. Therefore, using such an oracle is a natural choice to obtain reliable feedback in controllable environments.
>
> We would also like to clarify the role of this oracle. The interpreter uses pixel-level analysis only to determine whether the transition between two states is valid, but it does not provide any pixel-level supervision to the model. The model therefore receives feedback purely at the action level, which encourages it to learn actions that consistently lead to measurable progress rather than imitate oracle images.
>
> This point is also reflected in our empirical results. We observe that RL exhibits improved planning performance over SFT teacher forcing across all tasks (Table 1). It further generalizes better to larger out-of-distribution grid sizes (Table 9) and produces fewer invalid actions (Table 6). These findings suggest that the agent is not merely copying the oracle but is genuinely acquiring planning capability.
>
> [1] Daya Guo, Dejian Yang, Haowei Zhang, Junxiao Song, Ruoyu Zhang, Runxin Xu, Qihao Zhu, Shirong Ma, Peiyi Wang, Xiao Bi, et al. DeepSeek-R1: Incentivizing reasoning capability in LLMs via reinforcement learning. arXiv preprint arXiv:2501.12948, 2025.
>
> > A significant weakness in the method's detail is its reliance on non-learned, external modules to provide the reward signal.
>
> > the rule-based reward function (which relies on pixel comparison ) would be far too brittle to work.
>
> As the first work in visual planning, we adopt a rule-based state-action parsing function to provide a verifiable reward in a controlled setting. This component is only used for demonstration and can be replaced by alternative implementations of the interpreter. In more complex, structured environments, the reward signal can be provided by a learned reward model trained to evaluate visual state transitions directly, without relying on manually defined rules. But the core idea of the algorithm design remains unchanged, and we leave the implementation-wise choices for future research.

---

> ### Author Response · Authors · 2025-11-24
> **Response to Reviewer 2U49 by Authors (Part 2)**
>
> > Insufficient Justification for a "Purely Visual" Paradigm: This paper needs to justify why exploring a purely non-verbal, vision-only paradigm is a necessary or superior research direction.
>
> We would like to clarify that our intention is not to claim that non-verbal visual planning is superior to textual reasoning. Rather, our core contribution lies in demonstrating the possibility for planning in the visual domain that has not been previously proposed. We believe that exploring a purely visual non-verbal paradigm is meaningful for certain tasks where the underlying reasoning structure is inherently visual and/or "visual first" in nature, such as those involving spatial reasoning, geometric layouts, or state transitions that are more naturally captured through images. As typical spatial reasoning tasks, Maze, FrozenLake, and MiniBehavior fall into this category. In such settings, translating visual states into language may introduce unnecessary abstraction and errors, as we observe in FrozenLake (Fig. 4, 8, 9). In contrast, operating directly in the visual modality avoids this gap and can capture spatial and geometric relationships more naturally. Our empirical results further show that visual planning complements textual reasoning with improved performance (Table 1), indicating that **we use vision not for its own sake but because it provides an advantage in visual-first settings**. We believe that visual intelligence can contribute to future multimodal research by suggesting a novel dual-channel reasoning structure in MLLM, where visual and textual reasoning pathways can remain separate and interact only when needed.
>
> > Limited Task Complexity and Scalability: The paradigm is validated only on simple, 2D, discrete grid-world environments (FROZENLAKE, MAZE, MINIBEHAVIOR). It is highly questionable if this approach can scale to complex, 3D, photorealistic, or continuous-state environments (e.g., robotics).
>
> In our current settings, we choose grid-based environments for controllability, spanning different levels of pattern complexity (from simple Maze to visually richer FrozenLake) and action complexity (Maze and FrozenLake with 4 different actions and MiniBehavior with 6 actions). Experimental results show that our method performs well across these levels of complexity, illustrating its potential to be extended to more challenging environments. In this case, we believe our core reward design is broadly applicable to more complex planning tasks, as actions in most planning settings can naturally be categorized into one of three types (optimal, sub-optimal, and invalid). In our view, the main difficulty lies not in the reward formulation, but in defining reliable progress signals when visual transitions become more complex. This can be addressed by learning a reward model that evaluates state transitions directly from images, or by using trajectory-level rollouts, where the model uses final success signals to identify actions that lead to successful outcomes. We consider investigating this extension a natural next step and have highlighted this direction in the revised manuscript (L849-857).

---

> ### Author Response · Authors · 2025-11-24
> **Response to Reviewer 2U49 by Authors (Part 3)**
>
> > In such settings, autoregressively generating a perfect, step-by-step sequence of future images is computationally expensive (a point the authors concede)
>
> We agree that visual outputs introduce additional computational overhead relative to symbolic reasoning. However, we would also like to emphasize that this cost reflects a tradeoff for the better performance visual planning achieves in visual-centric tasks, where the textual reasoning paradigm falls short, even with fewer tokens. We also observe that thinking models, such as Gemini 2.5 Pro, produce the largest number of tokens among all tasks, indicating that visual planning is not always the most expensive option.
>
> To put this into context, we compare the average output token lengths of our visual planner with textual baselines, which is added in Appendix F.8. The tables below summarise the results across all tasks for the reviewer's convenience:
>
> | Model               | FrozenLake | Maze | MiniBehavior |
> |-------------------------------|-----------:|-----:|-------------:|
> | Gemini 2.0 Flash - Direct | 10.8      | 12.5 | 14.8 |
> | Gemini 2.0 Flash - CoT    | 150.5     | 166.5 | 196.5 |
> | Gemini 2.5 Pro (think)         | **885.6**  | **1030.2** | **1619.9** |
> | Qwen 2.5-VL-Instruct-7B -  Direct | 13.4 | 95.9 | 13.9 |
> | Qwen 2.5-VL-Instruct-7B - CoT    | 306.2 | 316.4 | 272.3 |
> | Qwen 2.5-VL-Instruct-7B - SFT (direct)   | 10.7 | 11.4 | 13.2 |
> | VPFT (ours)             |  819.2         | 957.2     | 1471.2 |
> | VPRL (ours)             |  819.2         | 957.2     | 1471.2 |
>
> We also include the cost of the trained textual planner variants on FrozenLake:
>
> | Model                    | FrozenLake |
> |----------------------------------|---------------------|
> | Qwen 2.5-VL-Instruct-7B - SFT (direct)                       | 10.7               |
> | Qwen 2.5-VL-Instruct-7B - SFT (w/ Coordinates)               | **179.0**              |
> | Qwen 2.5-VL-Instruct-7B - SFT (w/ ASCII)                     | 84.3               |
> | Qwen 2.5-VL-Instruct-7B - GRPO (w/ VPRL progress reward)      | 129.8              |
> | Qwen 2.5-VL-Instruct-7B - GRPO (w/ PR metric reward)          | 74.9               |
>
> These results show that visual planning does have a noticeable cost. However, the additional cost remains affordable when compared to textual CoT, suggesting that visual planning is still computationally feasible.
>
> As a complementary direction, we also advocate for further research into compact image representations using fewer tokens [2, 3], thereby reducing the computational cost of visualization generation.
>
> [2] Choudhury, R., Zhu, G., Liu, S., Niinuma, K., Kitani, K., and Jeni, L. Don’t look twice: Faster video transformers with run-length tokenization. Advances in Neural Information Processing Systems, 2024.
>
> [3] Qihang Yu, Mark Weber, Xueqing Deng, Xiaohui Shen, Daniel Cremers, and Liang-Chieh Chen. An image is worth 32 tokens for reconstruction and generation. Advances in Neural Information Processing Systems, 2024.

---

### Official Review · Reviewer_dg8a · 2025-11-01

**Soundness:** 3
**Presentation:** 3
**Contribution:** 3
**Rating:** 6
**Confidence:** 4

**Summary:**

This paper focuses on visual planning tasks in MLLMs. It proposes a novel way to perform visual planning by generating images during the planning process. Specifically, given an input image, the model reasons about the next state after performing certain actions by generating the corresponding images. To achieve this, the VLM is first trained to generate next-state images, then trained through reinforcement learning to encourage actions toward the target state. Experiments show that the proposed methods outperform strong open-source and private models in various environments.

**Strengths:**

1. Generating images makes the reasoning occur in the visual space rather than the textual space. Thus, the proposed method has the potential for more direct and better reasoning performance.
2. Through qualitative analysis of intermediate outputs, the paper shows that the proposed method can generate reasonable intermediate images, which is key for correct visual reasoning.
3. The experiment result demonstrates the proposed method outperforms strong baselines including private MLLMs.

**Weaknesses:**

1. It can be observed that the intermediate images are not perfect (e.g., in Fig. 3, first row, the player and goal tokens have artifacts). Thus, it would be interesting if the paper could show performance when the model reasons over high-quality images. For example, each time the model generates a new image, the corresponding high-quality image (rendered by the engine rather than generated by the model itself) is fed into the model. Would this lead to better performance? If so, the performance gap could quantify the importance of generating high-quality (precise) intermediate images.
2. If I understand correctly, $v$ stands for one of the intermediate images, as stated in L132–133. As such, how is it determined whether two images are an exact match in the EM metric in L300-L302? Or is it actually comparing in the action space (e.g., left/right) rather than the image space?
3. Similar to this work, some recent studies also explore generating intermediate images during reasoning, such as [1]. It is good that the paper discusses these related works, but it should consider directly comparing with these baselines.

[1] Imagine while Reasoning in Space: Multimodal Visualization-of-Thought

**Questions:**

Please check the weaknesses section:
1. (for weakness 1) How important it is to generate high-quality images? Would current intermediate images quality good enough?
2. (for weakness 2) If I understand correctly about EM measure, How to compute the exact match in the image space?
3. (for weakness 3) Comparison with similar methods such as MVoT?

---

> ### Author Response · Authors · 2025-11-24
> **Response to Reviewer dg8a by Authors (Part 1)**
>
> We thank the reviewer for their insightful comments. We appreciate the reviewer's positive remarks regarding the empirical strength of our paradigm and its potential. Below, we address the raised concerns about intermediate image quality and baseline comparisons in detail. We hope in light of the response, the reviewer could consider improving their score.
>
> > Would high-quality images rendered by the engine lead to better performance?
>
> We agree it would be valuable to analyze the impact of the intermediate image. To this end, we conducted an additional experiment on FrozenLake in Appendix F.7 to evaluate whether using high-quality images during inference improves performance. Specifically, we did not feed the model's self-generated image back into the input. We instead replaced it with the ground-truth image rendered by the program, which serves as the high-quality version. The results on FrozenLake are summarized below for the reviewer's convenience:
>
> ### Exact Match (EM, %) on FrozenLake under VPRL.
> | Model | L3 | L4 | L5 | L6 | Avg. |
> |----------------------------|------|--------|------|------|------|
> | VPRL (w/ self-generated images) | 97.6 | **95.6** | 90.8 | **82.4** | 91.6 |
> | VPRL (w/ ground-truth images) | **98.4** | 95.2 | **93.2** | 81.6 | **92.1** |
>
> ### Progress Rate (PR, %) on FrozenLake under VPRL.
> | Model | L3 | L4 | L5 | L6 | Avg. |
> |----------------------------|------|--------|------|------|------|
> | VPRL (w/ self-generated images) | 98.4 | **96.0** | 93.0 | **85.6** | 93.2 |
> | VPRL (w/ ground-truth images) | **98.5** | 95.8 | **94.1** | 85.3 | **93.4** |
>
> > (for weakness 1) How important it is to generate high-quality images?
>
> From the table, the performance with and without high-quality images remains very close across all grid sizes. This shows that our model is robust to visual noise and does not rely on perfectly rendered images to plan.
>
> > Would current intermediate images quality good enough?
>
> Although our work focuses on planning capability rather than image generation quality, the intermediate images produced by the model are sufficient for the planning tasks, which is confirmed by the table. We believe this behavior is encouraged by the dynamics interpreter. During the training, the dynamics interpreter serves as an implicit format constraint. Any generated image that cannot be parsed is treated as an invalid transition and receives a penalty, enforcing the model to maintain the semantic structure of the environment in its generated images.
>
> > It can be observed that the intermediate images are not perfect (e.g., in Fig. 3, first row, the player and goal tokens have artifacts).
>
> We would like to clarify that these artifacts are more likely to come from the limitation of the image tokenizer, rather than from the model’s image generation ability. Even when we encode a ground-truth image into tokens and then decode it back, the tokenizer inevitably introduces noise. As a result, the reconstructed image is not identical to the original and contains similar artifacts as the reviewer observed in Fig.3. We have added Appendix F.7 to provide a clearer comparison between the original images and the reconstructed outputs (Figure 16).

---

> ### Author Response · Authors · 2025-11-24
> **Response to Reviewer dg8a by Authors (Part 2)**
>
> > If I understand correctly, stands for one of the intermediate images, as stated in L132–133. As such, how is it determined whether two images are an exact match in the EM metric in L300-L302? Or is it actually comparing in the action space (e.g., left/right) rather than the image space?
>
> We would like to clarify that the "exact match" refers to equivalence at the **action level** rather than a pixel-wise match between images. Starting from the input state $v_0$, we check each subsequent state by verifying whether it can be reached from the previous state using the same action. In other words, two states are considered an exact match if they represent the same environment configuration, even if their pixel values are not identical. We have updated the revised manuscript accordingly in L302–306.
>
> > (for weakness 2) If I understand correctly about EM measure, How to compute the exact match in the image space?
>
> For the EM measure, we rely on the dynamics interpreter to parse each state and extract the underlying action. Specifically, it identifies the agent's position and checks whether the other objects in the environment remain consistent through a pixel-wise feature extractor that divides the images into grid cells. Based on the change in configuration, the interpreter then determines both the action type (e.g., up) and the validity of the transition. After parsing both the predicted trajectory and the optimal trajectory into action sequences, we compare these two sequences directly to compute the metric. Further details of the interpreter are provided in Appendix E.3.
>
> > (for weakness 3) Comparison with similar methods such as MVoT?
>
> While both methods generate intermediate images, we would like to clarify that MVoT adopts a different reasoning paradigm from ours. MVoT still performs its reasoning extensively in the textual domain. The images it produces act as visualizations of its textual reasoning, and such visualizations serve mainly as supplementary information. In contrast, our visual planner represents the environment, selects actions, and updates the state entirely through visual representation. Since these two approaches rely on different forms of mediation and assign different roles to their intermediate images, a direct comparison would not be well aligned with the scope of our work.
>
>
> That said, we greatly value the reviewer's suggestion, and we agree that connecting these two directions would be helpful. Treating MVoT as an additional textual variant would allow us to further examine the advantages of visual planning. However, we are unable to include this study at this stage due to time constraints, but we see it as a meaningful extension and will examine it in future work.

---

### Author Response · Authors · 2025-12-03
**Summary of the Rebuttal**

We sincerely thank the reviewers for the time spent evaluating our work and for the constructive feedback. Based on the comments, we have revised the manuscript. Below, we summarize the key changes.

## Experiments on Computational Cost:
We appreciate the question about computational cost raised by reviewers 2U49 and Rhdx.  We agree that this is a very valuable point. Hence, we have computed the average output token number for our visual planning models and textual baselines. As a common question, we summarize the result here for the reviewers’ convenience and kindly refer the reviewer to Appendix F.8 for the detailed result and analysis.

| Model               | FrozenLake | Maze | MiniBehavior |
|-------------------------------|-----------:|-----:|-------------:|
| Gemini 2.0 Flash - Direct | 10.8      | 12.5 | 14.8 |
| Gemini 2.0 Flash - CoT    | 150.5     | 166.5 | 196.5 |
| Gemini 2.5 Pro (think)         | **885.6**  | **1030.2** | **1619.9** |
| Qwen 2.5-VL-Instruct-7B -  Direct | 13.4 | 95.9 | 13.9 |
| Qwen 2.5-VL-Instruct-7B - CoT    | 306.2 | 316.4 | 272.3 |
| Qwen 2.5-VL-Instruct-7B - SFT (direct)   | 10.7 | 11.4 | 13.2 |
| VPFT (ours)             |  819.2         | 957.2     | 1471.2 |
| VPRL (ours)             |  819.2         | 957.2     | 1471.2 |

We observe that thinking models, such as Gemini 2.5 Pro, produce the largest number of tokens among all tasks, indicating that visual planning is not always the most expensive option. We would also like to emphasize that this cost reflects a tradeoff for the better performance visual planning (VPRL with 80.6% average EM in Table 1) achieves in visual-centric tasks, where the textual reasoning paradigm falls short (Gemini 2.5 Pro with 43.7% average  EM), even with fewer tokens (trained Qwen 2.5-VL-Instruct-7B with 53.6% average EM).

## Key Changes

Here we summarize the key changes in the updated manuscript. All updated parts are highlighted in $ \textcolor{magenta}{magenta} $ for the convenience of the reviewers.

- We added an explanation of how state equality is defined in Section 3.
- We updated the limitation section in Appendix D to reflect the direction of scaling the framework to more complex environments with either learned reward models or final-success rewards.
- We added an analysis of image quality in Appendix F.7.
- We added a computational cost analysis in Appendix F.8.

## Conclusion

We thank the positive cognition from reviewers dg8a, uu3p, and Rhdx, including the novelty of visual planning and the strength of our experiment results. We also thank reviewer 2U49 for the insightful comments and for noting that our paradigm is *genuinely new* and *well-designed*. We are grateful for the reviewer 2U49's openness to raising the score and have responded in detail to the concerns regarding reward design, computational cost, and the motivation of our work. We believe our response fully addresses these concerns, although there is no discussion in the rebuttal period due to the recent disruptions.

We thank all reviewers again for the time and effort dedicated to reviewing our submission.

Best,

Authors

---

### Public Comment · ~Weiyu_Ma1 · 2026-03-15
**Question: Relationship to Visual MPC and World Models (Dreamer, PlaNet)**

Dear Authors,

Congratulations on the oral. The paper is well-written and I find the direction interesting. I have one main question regarding the positioning of the contribution.

The idea of planning directly in visual/perceptual space has been explored extensively in model-based RL. Visual Foresight / Visual MPC (Finn & Levine, 2017; Ebert et al., 2018) predicts future image frames and optimizes actions in pixel space. The Dreamer family (Hafner et al., 2019–2023) performs imagination rollouts in latent space — adding a decoder would yield a pipeline conceptually very close to this work: generating visual state sequences for planning with agent-produced actions. Could the authors comment on the fundamental distinction from these approaches, and why they were not discussed or included as baselines?

A related concern is that the evaluation environments (FrozenLake, Maze, MiniBehavior) are discrete grid-worlds where the image is nearly a direct rendering of the symbolic state. In this setting, visual representations outperforming textual descriptions seems somewhat expected, since text introduces lossy spatial abstraction. Comparing only against text-based methods makes it hard to tell whether the gains come from the proposed paradigm itself or simply from avoiding language — something the model-based RL literature established long ago.

Best regards

---

### Meta-Review · Area_Chair_TMRg · 2026-01-09

**Summary:**

This submission proposes Visual Planning, a novel paradigm for reasoning through purely visual representations, complementing language-based reasoning for "vision-first" tasks (e.g., spatial/geometric planning). Reviewers uniformly recognized the work’s novelty, strong empirical results, and methodological robustness. Key concerns raised included: (1) intermediate image quality and the EM metric’s definition (dg8a); (2) reliance on external oracles for rewards, justification of the "purely visual" paradigm, task scalability, and computational cost (2U49); (3) need for explicit comparisons between language/visual media, transferability to 3D/real CV tasks, and integration with MLLMs (uu3P); (4) limited task scope (discrete grid-worlds), lack of explicit action sequences for debugging, and computational efficiency relative to textual CoT (Rhdx). The authors’ comprehensive rebuttal addressed nearly all core concerns with additional experiments, clarifications, and future directions, reinforcing the work’s validity and promise.

**Reviewer Concerns:**

### Addressed Concerns
- Reviewer dg8a: (1) Intermediate image quality: Authors demonstrated robustness via experiments replacing self-generated images with ground-truth (minimal performance gain, Appendix F.7); (2) EM metric clarification: Confirmed it evaluates action-level equivalence (not pixel-wise) via dynamics interpreter parsing; (3) MVoT comparison: Explained paradigm differences (MVoT uses images as textual supplements, Visual Planning is purely visual) and noted future work.
- Reviewer 2U49: (1) External oracle reliance: Aligned with RLVR best practices, clarified the oracle provides action-level feedback (not pixel supervision) and model learns planning, not imitation; (2) "Purely visual" justification: Emphasized complementarity (not superiority) for visual-first tasks, avoiding language abstraction errors; (3) Computational cost: Provided token count comparisons showing visual planning’s cost is a reasonable tradeoff for better performance (Appendix F.8); (4) Scalability: Outlined learned reward models and trajectory-level rollouts for complex environments.
- Reviewer uu3P: (1) Language vs. visual medium figures: Referred to Figures 4, 8, 9 (textual models’ spatial errors); (2) 3D transfer: Core reward design (optimal/sub-optimal/invalid actions) is applicable, with future work planned; (3) GRPO + MLLM integration: Proposed dual-channel reasoning (visual/textual pathways) and training strategies; (4) Real CV task advantages: Visual representations retain spatial/geometric structure critical for CV tasks.
- Reviewer Rhdx: (1) Computational efficiency: Token count data showed visual planning is feasible relative to textual CoT (e.g., Gemini 2.5 Pro’s higher token output); (2) Task scope: Validated feasibility in controlled grid-worlds (varying complexity) with clear extension paths; (3) Interpretability: Visual state sequences provide traceable decision paths for debugging.

### Outstanding Concerns
- Reviewer dg8a: Direct empirical comparison with MVoT (authors cite time constraints, defer to future work).
- Reviewer 2U49: Empirical validation of scalability to 3D/photorealistic environments (authors outline directions but no current experiments).

**Reviewer Scores:**

- Reviewer dg8a: 6 -> 8. The rebuttal fully addressed image quality, EM metric, and MVoT clarification, reinforcing the work’s strengths.
- Reviewer 2U49: 4 -> 6 The authors resolved core concerns (reward design, paradigm justification, computational cost) as the reviewer indicated they would raise the score if satisfied.
- Reviewer uu3P: 6 -> 8. The rebuttal addressed all questions clearly, with strong presentation and novel contributions retaining their value.
- Reviewer Rhdx: 8 -> 8. The rebuttal confirmed computational feasibility and interpretability, with no concerns undermining the work’s quality.

---

### Decision · Program_Chairs · 2026-01-26

Accept (Oral)